# Prenatal and postnatal determinants of stunting at age 0–11 months: A cross-sectional study in Indonesia

**Arindah Nur Sartika[1]¤, Meirina Khoirunnisa[1], Eflita Meiyetriani❶[2]\*, Evi Ermayani[2], Indriya Laras Pramesthi[2], Aziz Jati Nur Ananda❶[2]**

**1** Department of Nutrition, Faculty of Medicine Universitas Indonesia, Jakarta, Indonesia, **2** SEAMEO Regional Center for Food and Nutrition, Pusat Kajian Gizi Regional Universitas Indonesia, Jakarta, Indonesia

¤ Current address: Mitra Keluarga School of Health Sciences, East Bekasi, Indonesia
\* eflita@seameo-recfon.org

## Abstract

### Background

Childhood stunting remains the most challenging consequence of undernutrition because it is associated with suboptimal brain development and the subsequent increased risk of metabolic diseases and decreased earnings in adult life. The Sambas District in Indonesia has a high prevalence of stunting (28.4%), followed by underweight (25.5.%) and wasting (14.4%) among children, as well chronic energy deficiency (27%) and anemia (62%) among pregnant women. The present study sought to determine significant factors in childhood stunting with a focus on maternal and child nutrition and prenatal and postnatal determinants.

### Methods

This prospective, repeat, cross-sectional study investigated factors associated with stunting among 559 infants age 0–11 months in Sambas District, Indonesia. Anthropometric measurements were performed by trained enumerators. Data from a 2016 survey of pregnant women and a 2017 survey on mothers and their children were used for postnatal data collection to quantify the prevalence of stunting at age 0–11 months. Using 20 potential predictors of stunting categorized by household characteristics, maternal characteristics, antenatal care services, and child characteristics, logistic regression analysis was conducted to assess the adjusted association between stunting and these factors.

### Results

Of 559 children analyzed, 20.8% were stunted. In the model with low birth weight (LBW) as predictor for stunting, the odds of stunting increased significantly among children who weighed <2.500 g at birth; children who had diarrhea in the past 2 weeks and children who had incomplete basic immunization coverage as infants age 9–11 months. In model without LBW, the odds of stunting increased significantly among children who had preterm at birth, short maternal stature and children who had incomplete basic immunization coverage for 9–11 months infants.

**Data Availability Statement:** All relevant data are within the paper and its Supporting information files.

**Funding:** This study was supported by grants from the Indonesian Ministry of Education and Culture through Southeast Asian Ministers of Education Organization Regional Centre for Food and Nutrition (SEAMEO RECFON) for the data collection, data management, data analysis and manuscript writing (Number: 056A/RECFON-SK/III/2017). This publication was funded by the Directorate of Research and Community Engagement of Universitas Indonesia (Number: NKB-3049/UN2.R3.1/HKP05.00/2019). Indonesian Ministry of Education and Culture and Directorate of Research and Community Engagement of Universitas Indonesia had no role in study design, data collection and analysis, decision to publish, or preparation of the manuscript.

**Competing interests:** The authors have declared that no competing interests exist.

**Abbreviations:** CI, confidence interval; LBW, low birth weight; OR, odds ratio; WHO, World Health Organization.

## Conclusions

Postnatal factors—preterm birth, low birth weight, diarrhea and complete basic immunization coverage—were associated with infant stunting in Sambas District, Indonesia. The prenatal factors such as short maternal stature were significant in the multivariate model. Policy makers, especially in the government, should recommend measures focused on those prenatal and postnatal factors to prevent stunting in children and to avoid the sequelae of childhood stunting in adult life.

## Introduction

Childhood stunting remains one of the most fundamental challenges to overcome in human development. Stunting is associated with suboptimal brain development, which has long-lasting consequences with regard to cognitive ability, school performance, and future earnings as an adult [1]. An estimated 162 million children are stunted worldwide, of whom 56% live in Asia and 36% in Africa [2].

In Indonesia, childhood stunting remains at a high level and continues to be a serious public health problem, for which progress in reducing childhood undernutrition has been slow over the past decade. The trend from the Indonesia Family Life Survey from 1997 to 2007 indicates a reduction in stunting among children age <5 years in Indonesia of only 10.1% (a decrease from 46.8% to 36.7%) [3]. These survey data mirror the 2007 findings of the Basic Health Research Survey, which reported that approximately 36.8% of children age <5 years in Indonesia were stunted. Based on 2013 data from the Basic Health Research Survey, the current proportion of children with stunting in Indonesia is high and has increased to 37.2% [4].

Nutritional status, especially in pregnant women and children age <2 years, is a significant problem in some areas of Indonesia, including Sambas District, West Kalimantan Province. This area is known to have both a high prevalence of stunting in children age <5 years and of chronic energy deficiency in pregnant women. The 2013 Basic Health Research Survey reported 51.4% of children age <5 years had stunting, 34.1% were underweight, and 21.7% had wasting.

In 2016, a cross-sectional study was conducted in Sambas District to assess the nutritional status of children age <2 years and pregnant women and the associated factors. The study evaluated 633 children age <2 years and 551 pregnant women across 30 villages from 19 subdistricts, including a total of 1184 households [5]. The prevalence of stunting was 28.4%, underweight was 25.5.%, and wasting was 14.4% among the children, whereas the prevalence of chronic energy deficiency among pregnant women was 27%, with a median upper arm circumference of 25.2 cm. Moreover, the prevalence of anemia among pregnant women was 62% with a median hemoglobin of 10.6 g/dL. Based on gestational age, almost 50% of anemic pregnant women (mild, moderate, and severe anemia) were in their second trimester. Those situations were categorized as significant public health problems based on World Health Organization (WHO) criteria.

Based on all of these findings, the present author research team was interested in following up the outcomes for the 2016 cross-sectional study participants[5], especially those who were pregnant at that time, and the health outcomes of the infants born in 2017. The present study aimed to assess associations between pre- and postnatal conditions and stunting among infants age 0–11 months in Sambas District, Indonesia.

According to several studies, stunting results from a chronic process that spans from pregnancy into the early life of the infant. No single factor has been identified as a determinant of this condition, including prenatal, birth conditions, and postnatal factors [6]. To determine the significant factors that contribute to stunting, a study must observe the relationships between these factors from pregnancy through early life of the infant.

## Materials and methods

### Area, time, and data source

This study was conducted in Sambas District, which has total area about 6.395,70 km$^2$, comprising 4.36% of the total area of West Kalimantan Province. Sambas District is classified as a rural and border area of Indonesia and Malaysia. Study participants resided in 17 of the 19 subdistricts within 30 villages throughout the district.

The data for this study were derived from two cross-sectional studies in 2016 and 2017. The data management for 2016 was conducted from April–August 2016 over several successive phases: (1) preparation from April 11–24; (2) data collection from April 25–May 11; (3) data entry, data cleaning, and analysis in June 2016; and (4) the report writing phase through August 2016.

This follow-up study in 2017 was conducted from April–June 2017 over several phases: (1) preparation from April 3–14; (2) data collection May 17–April 2; (3) data entry and data cleaning and analysis in May 2017; and (4) the report writing phase in late May–June 8, 2017.

### Study design

This study was designed as a repeated cross-sectional study of child birth outcomes and nutritional status related to the conditions during pregnancy determined from a previous study in Sambas District. The report focused on the prenatal, perinatal, and postnatal factors and their association with stunting in infants age 0–11 months. This study is part of a larger study, "Association Between Maternal Nutritional Status During Pregnancy and Birth Outcome and Nutritional Status of Their Infant Age 0–11 Months in Sambas District, West Kalimantan Province" conducted by South East Asian Ministers of Education Organization (SEAMEO) Regional Center for Food and Nutrition (RECFON) and Department of Nutrition, Faculty of Medicine, Universitas Indonesia.

### Ethics statement

This study was conducted after receiving ethical clearance from the Ethical Committee of Faculty of Medicine of the University of Indonesia (Number 329/UN2.F1/ETIK/2016, April 25, 2016). Permission from the local government (provincial and district) as well as the local health authority were also obtained prior to conducting this study. To follow the basic principles of bioethical research, the survey team explained the details of the study, emphasized the voluntary nature of the participation, collected data safely, and ensured confidentiality of the data. Written consent was obtained from study participants.

### Study population and subjects

The target population for the study was mothers who joined the previous study in April 2016 during their pregnancy. The study participants were both the mothers of caregivers of the infants and the infants whose mothers participated in the previous study. Inclusion criteria were Sambas District residence; being either the mother or the caregiver of the infant or being the infant whose mother was interviewed in the previous study; and willingness to participate

in the study. Exclusion criteria were current residence of the potential participant in an area outside Sambas District and inability to be contacted.

## Sample size calculation and sampling procedure

**Sample size.** The sample size was calculated based on supporting information from a previous study in Sambas District in 2016 [5]. A total of 559 participants was the minimum required sample to conduct the follow-up study.

**Sampling procedure.** In Sambas District, 30 villages from the 183 villages in this district were chosen randomly. The minimum proportionate number of pregnant mothers from each village to be recruited was achieved by random sampling using probability proportional to size. From the list of pregnant women in the 2016 study, confirmations were made with local midwives to determine the participants who were lost to follow-up; then the research team followed up with women who remained on the list.

## Data collection procedure

**Structured interview.** The interview was performed using a structured questionnaire that was tested before data collection. The oral interview was conducted over 30–60 minutes with the mothers or caregivers. To help facilitate the interview, enumerators also asked participants to bring their Health of Mother and Child Book and family certificate. However, to obtain complete data, enumerators also asked key informants such as midwives or other family members who may better understand the child's condition if caregivers did not understand or remember, such as date of birth, birth weight, and gestational age when the mother delivered the infant. Before starting data collection, mothers received information about the study and asked if they wanted to participate. If they were willing to join, they signed the informed consent document; if they were illiterate, they signed using a finger stamp and they were assisted by their relatives or regional health officers or cadre.

This study used several strategies to reduce information bias. Interview guidelines were created for the enumerators, and a pretesting questionnaire was administered before conducting the study. During the interview, the respondents were asked to provide some evidence regarding their answers. For example, for immunization coverage, participants were asked to show their Health of the Mother and Child Book. Similarly for birth weight, participants were asked to provide the birth certificate and also the record of the child's weight history.

## Variables and operational definition

In this study, the prenatal and postnatal variables were considered explanatory variables, for which the corresponding coding definitions are shown in Table 1.

## Anthropometric measurements for children and mothers

To assess the nutritional status of the child, anthropometric data were collected by measuring the child's weight and length. Child length was measured using a length board, and weight was measured by weight scale (SECA). Date of birth was input by asking the mother and checking in the Health of Mother and Child Book. Both length and weight were assessed twice, and then the average was noted. The procedure for performing anthropometric measurements was used as specified by WHO. The measurements were conducted from May 17–April 2, 2017.

To assess the nutritional status of the mother, anthropometric data were collected by measuring weight and height. Height was measured using length board, and mother's weight were

**Table 1. Variables and operational definition.**

| Category | Variable | Definition | Time of study |
|---|---|---|---|
| **Nutritional Status** [7] | Underweight (infant) | Nutritional status based on weight for age <−2 standard deviation (SD) of the WHO Child Growth Standards median | 2017 |
| | Stunting (infant) | Nutritional status based on length for age <−2 SD of the WHO Child Growth Standards median | 2017 |
| | Wasting (infant) | Nutritional status based on weight for length <−2 SD of the WHO Child Growth Standards median | 2017 |
| **Prenatal factors** | Maternal Anaemia during pregnancy | Haemoglobin level below 11 mg/dl in pregnant women | 2016 |
| | Maternal Short Stature [8] | A maternal height of ≤145 cm | 2016 |
| | Maternal CED risk during pregnancy | Maternal risk of Chronic Energy Deficiency (CED) shown by a MUAC (Mid-Upper Arm Circumference) measurement. Mid-Upper Arm Circumference (MUAC)<br>• CED risk <23,5 cm<br>• Non CED risk ≥23,5 cm | 2016 |
| | **Knowledge** | | 2016 |
| | Knowledge during pregnancy | Cognitive ability to know and understand issues related to nutrition and health measured during pregnancy. Cut off based on median value. | |
| | Knowledge on IYCF | Cognitive ability to know and understand issues related to Infant and Young Child Feeding (IYCF) measured after giving birth. Cut off based on median value. | 2016 |
| | Knowledge on Stunting | Cognitive ability to know and understand issues related to Stunting measured after giving birth. Cut off based on median value. | 2016 |
| **Postnatal factors** | | | |
| **Morbidity** | Diarrhea | When three or more stools are passed in 24 hours that are sufficiently liquid to take the shape of the container in which they are placed in the last 2 (two) weeks. Presence and how long occurrence of diarrhea in the last two weeks | 2017 |
| | Acute Respiratory Infection (ARI) | Infectious disease by virus or bacteria, begin with fever accompanied by one or more of these symptoms: sore throat, pain swallow, hacking cough, or cough with phlegm. Presence how long occurrence of ARI in the last two weeks | 2017 |
| **Birth Condition** | Preterm birth | Babies born alive before 37 weeks of pregnancy are completed | 2017 |
| | Low birth weight [9] | Low birth weight has been defined by WHO as weight at birth of <2500 grams (5.5 pounds) | |
| **Utilization of health care service** | Health care services utilization | The utilization of available health care services.<br>Complete: ≥4 visits<br>Incomplete: <4 visits | 2017 |
| | Ever immunized | Number of children ever immunized at least once | 2017 |
| | Neonatal visit | Visit to health care provider during the first 6–48 hours (KN1), 3–7 days (KN2), and 8–28 days (KN3) of the neonatal birth. | 2017 |
| | Vitamin A supplementation coverage | Receiving vitamin A capsule every 6 months. (February and August) for infant ≥6 months old (Red capsule (100.000 IU) for 6–11 months) | 2017 |
| | Complete immunization coverage children 9–11 months | Children who had complete basic immunization based on Indonesian Government regulation | 2017 |
| | Household food security | A state that exists when all people at all times have both physical and economic access to sufficient food to meet their dietary needs for a productive and healthy life. US-Household Food Security Survey Module (US-FSSM)<br>Category for household with one or more child present: • Food secure: 0–2, •Food insecure: 2–18 | 2017 |

CED = chronic energy deficiency; WHO, World Health Organization.

measure by weight scale (SECA), and date of birth were input by asking the mother and cross-referencing with the family certificate. Both the mother's height and weight were assessed twice, and then the average was recorded. The measurements were conducted in the previous study from April 25–May 11, 2016.

## Statistical analysis

Descriptive analysis was performed on participant's characteristics and variables included in this study. Categorical data were applied to display the frequency and percentage. Numeric data were also reported as mean scores and standard deviations to determine the collected data were normally distributed. If the collected data were not normally distributed, they were presented as median, minimum, maximum value and interquartile range. To analyze normal distribution of data, the Kolmogorov–Smirnov Z test was used.

With regard to the association between several variables and stunting, bivariate analysis was performed using the chi-squared or Fisher's exact test. The pre- and postnatal determinants of stunting were determined to be contributing factors after assessment in multivariable analysis using multiple logistic regressions. Potential confounders were first selected on the basis of their known association with child stunting in developing countries. For all tests, $p < 0.05$ was considered statistically significant. The odds ratios with 95% CIs were calculated in order to assess the adjusted risk of independent variables. All analyses were performed using SPSS (version 26.0, IBM, Armonk, NY, USA). For missing data, complete case analysis (listwise deletion) was used. With this approach, a case is dropped from an analysis because it has a missing value in at least one of the specified variables. The analysis is only run-on cases which have a complete set of data.

# Results

Of the 559 respondents from the 2016 study, 500 were followed up in the present study. Of the original 559 participants, 59 were lost to follow-up for these reasons: 26 (44%) moved or worked outside Sambas District, 19 (32%) could not be contacted, 9 (15%) moved or worked abroad, and 5 (9%) refused to participate. Of the 500 potential participants, 16 reported infant mortality for these reasons: 7 (43.8%) due to an abnormality, 2 (12.5%) due to infectious disease, and 7 due to other reasons. The final sample size for analysis was 484 caregivers–child pairs (Table 2).

Table 2 provides data on the socioeconomic characteristics of the mothers of children age 0–11 months and their children. Data showed 45% of the children were age >9 months, 51.7% of the mothers had a nuclear family, and 51.1% of the mothers had <9 years of schooling. Regarding the household characteristics, >90% of households (93.5%) had Melayu ethnicity and had a father as the head of family. Almost 98% of households reported children age < 2 years in the household, and only 1.4% had 2 children age 2–5 years in the household. Regarding food security, the percentage of households falling within the insecure category was high, at 58.7%, and >50% of households had a monthly income <2,000,000 Indonesian rupiahs in the past month.

## Bivariate and multivariable analysis for association of pre- and postnatal determinants of stunting

As shown in Tables 3–5, bivariate analysis results showed that short maternal stature, preterm birth, low birth weight (LBW), children age ≥6 months, diarrhea, complete immunization coverage, income per capita below the median, and poor knowledge of infant and young child feeding were associated with stunting in infants.

Multivariable logistics regression was adjusted for sociodemographic variables including sex and age of the children. The results of multiple logistic regression models for prenatal and postnatal factors revealed that prenatal factors which was maternal stature and post-natal

**Table 2. Sociodemographic characteristics and economic conditions of study participants.**

| Variables | n | % |
|---|---|---|
| **Sociodemographic characteristics of the study participants (N = 586)** | | |
| Child age (N = 484) | | |
| 0–5 months | 59 | 12.2 |
| 6–8 months | 207 | 42.8 |
| ≥9 months | 218 | 45 |
| Type of family (N = 484) | | |
| Nuclear | 250 | 51.7 |
| Extended | 234 | 48.3 |
| Head of household (N = 484) | | |
| Mother | 4 | 0.8 |
| Father | 454 | 93.8 |
| Grandparents | 24 | 5 |
| Other | 2 | 0.4 |
| Mother's educational level* (N = 556) | | |
| No schooling | 26 | 4.7 |
| <9 years of schooling | 284 | 51.1 |
| ≥9 years of schooling | 246 | 44.2 |
| Ethnicity* (N = 557) | | |
| Melayu | 521 | 93.5 |
| Dayaknese | 23 | 4.1 |
| Javanese, Bugisnese, Chinese and Maduranese and other | 13 | 2.4 |
| Number of under-two children in the household (N = 484) | | |
| 1 child | 474 | 97.9 |
| 2 children | 10 | 2.1 |
| Number of 2–5 years old children in the household (N = 484) | | |
| 0 child | 360 | 74.4 |
| 1 child | 116 | 20.8 |
| 2 children | 8 | 1.4 |
| **Economic Condition** | | |
| Household food security (N = 484) | | |
| Insecure | 284 | 58.7 |
| Secure | 200 | 41.3 |
| Household income last month (N = 553) | | |
| ≥2,000,000* | 253 | 45.8 |
| <2,000,000* | 300 | 54.2 |
| Household expenditure last month (N = 428) | | |
| ≥1,561,500 IDR | 214 | 50 |
| <1,561,500 IDR | 214 | 50 |
| Income per capita | | |
| Median, Min-Max, IQR | IDR 377.500, IDR 20.000–380.000, IDR 400.000 | |

IDR = Indonesian rupiah.

factors including preterm birth and complete immunization coverage were significantly related to stunting.

In the model without LBW included, children with preterm birth had higher odds of being stunted (aOR = 5.10; 95% CI: 1.04–24.8) compared to children with term birth (born

**Table 3. Child characteristics and stunting.**

| Variable | Total | Stunting | COR (95%CI) | P value | AOR (95%CI) [a] | P value [b] | AOR (95%CI) [c] | P value [d] |
|---|---|---|---|---|---|---|---|---|
| Child Age[#] (N = 481) | | | | | | | | |
| ≥6 months | 422 | 95 (22.5) | 3.14 (1.2–8.17) | **0.013***  | - | - | - | - |
| <6 months | 59 | 5 (8.5) | 1 | | - | - | - | - |
| Child sex | | | | | | | | |
| Female | 217 | 40 (18.4) | 0.77 (0.49–1.20) | 0.248 | 0.99 (0.45–2.16) | 0.973 | 0.809 (0.462--1.416) | 0.459 |
| Male | 264 | 60 (22.7) | 1 | | 1 | | 1 | |
| Low birth weight (N = 480) | | | | | | | | |
| Yes | 36 | 19 (52.8) | 5.01(2.49–10.06) | **<0.001***  | - | - | 4.119 (1.712--9.91) | **0.002*** |
| No | 444 | 81 (18.2) | 1 | | - | - | 1 | |
| Preterm birth (N = 481) | | | | | | | | |
| Yes | 38 | 13 (34.2) | 2.13 (1.05–4.33) | **0.034***  | 5.10 (1.04–24.8) | **0.044** | 2.514 (0.959--6.588) | 0.061 |
| No | 443 | 87 (19.6) | 1 | | | | 1 | |
| Diarrhea in the last 2 weeks (N = 480) | | | | | | | | |
| Yes | 64 | 25 (39.1) | 2.96 (1.69–5.19) | **<0.001***  | 2.79 (0.99–7.90) | 0.053 | 3.277 (1.615--6.647) | **0.001*** |
| No | 416 | 74 (17.8) | 1 | | 1 | | 1 | |
| Complete immunization children 9–11 months (N = 206) | | | | | | | | |
| Incomplete | 79 | 27 (34.2) | 1.92 (1.02–3.61) | **0.040***  | 2.65 (1.14–6.18) | **0.024** | 2.43 (1.028--5.761) | **0.043*** |
| Complete | 127 | 27 (21.3) | 1 | | 1 | | 1 | |
| Child Vitamin A supplementation (N = 416) | | | | | | | | |
| Yes | 304 | 73 (24) | 0.77 (0.40–1.19) | 0.181 | 0.72 (0.22–2.71) | 0.682 | 0.725 (0.199--2.628) | 0.624 |
| No | 112 | 20 (17.9) | 1 | | 1 | | 1 | |
| Neonatal Examination (N = 478) | | | | | | | | |
| Complete (3 times) | 148 | 36 (24.3) | 0.73 (0.46–1.17) | 0.192 | 0.85 (0.36–2.04) | 0.724 | 0.849 (0.353--2.04) | 0.715 |
| Incomplete (<3 times) | 330 | 63 (19.1) | 1 | | 1 | | 1 | |

Note: COR: Crude Odds Ratio, AORa = Adjusted Odds Ratio without LBW, P value b = P value without LBW, AORc = Adjusted Odds Ratio with LBW, P value d = P value with LBW, CI = Confidence Interval, adjusted variables were child sex, preterm birth, diarrhea in the last 2 weeks, complete immunization children, child vitamin A supplementation and neonatal examination.

alive after 37 weeks of pregnancy are completed). The odds of being stunted were higher (aOR = 3.28; 95% CI: 1.61–6.65) among children with incomplete immunization coverage compared to children who had complete immunization coverage (aOR = 2.65; 95% CI: 1.14–6.18).

Children with LBW had higher odds of being stunted (adjusted odds ratio [aOR], 4.12; 95% confidence interval [CI], 1.71–9.91) compared with children with normal birth weight. The odds of being stunted were higher (aOR, 3.28; 95% CI, 1.61–6.65) among children with diarrhea in the past 2 weeks compared with children without diarrhea. Children who had incomplete immunization coverage were likely to be stunted compared with children who had complete immunization coverage (aOR, 2.43; 95% CI, 1.03–5.76).

**Table 4. Mother/caregiver characteristics and stunting.**

| Variable | Total | Stunting | COR (95%CI) | P values (95%CI) | AOR (95%CI) [a] | P value [b] | AOR (95%CI) [c] | P value [d] |
|---|---|---|---|---|---|---|---|---|
| Age of childbearing** (N = 479) | | | | | | | | |
| <18 years old | 12 | 3 (25) | 1.28 (0.34–4.81) | 0.720 | 1.67(0.25–11.08) | 0.594 | 1.99 (0.298-–13.27) | 0.538 |
| ≥18 years old | 469 | 97 (20.7) | 1 | | 1 | | 1 | |
| Maternal CED risk during pregnancy** (N = 481) | | | | | | | | |
| No | 349 | 73 (20.9) | 0.97 (0.59–1.60) | 0.911 | 1.15 (0.50–2.69) | 0.734 | 1.12 (0.468-–2.68) | 0.799 |
| Yes | 132 | 27 (20.5) | 1 | | 1 | | 1 | |
| Maternal Anemia during pregnancy** (N = 481) | | | | | | | | |
| No | 185 | 43 (23.2) | 0.79 (0.50–1.23) | 0.295 | 0.65 (0.28–1.48) | 0.305 | 0.611 (0.348-–1.07) | 0.086 |
| Yes | 296 | 57 (19.3) | 1 | | 1 | | 1 | |
| Maternal dietary diversity** (N = 481) | | | | | | | | |
| Not diverse | 289 | 65 (22.5) | 1.302 (0.82–2.06) | 0.259 | 0.99(0.20–4.79) | 0.989 | 0.918 (0.277-–3.04) | 0.073 |
| Diverse | 192 | 35 (18.2) | 1 | | 1 | | 1 | |
| Maternal stature (N = 476) | | | | | | | | |
| <145 cm | 91 | 31 (34.1) | 2.41 (1.45–4.00) | 0.001* | 2.48 (1.05–5.84) | **0.038** | 1.77 (0.933-–3.37) | 0.890 |
| ≥145 cm | 385 | 68 (17.7) | 1 | | 1 | | 1 | |
| Frequency of ANC visit** (N = 481) | | | | | | | | |
| Complete | 347 | 75 (21.6) | 0.83 (0.50–1.38) | 0.474 | 1.17 (0.22–4.58) | 0.775 | 0.910 (0.476-–1.736) | 0.776 |
| Incomplete | 134 | 25 (18.7) | 1 | | 1 | | 1 | |
| Knowledge during pregnancy** (N = 481) | | | | | | | | |
| Good knowledge | 445 | 93 (20.9) | 0.91 (0.39–2.15) | 0.836 | 1.01 (0.3–2.50) | 0.992 | 0.862 (0.298–2.499_ | 0.785 |
| Poor knowledge | 36 | 7 (19.4) | 1 | | 1 | | 1 | |
| Knowledge about IYCF (N = 481) | | | | | | | | |
| Poor knowledge | 344 | 82 (23.8) | 2.07 (1.19–3.60) | 0.009* | 2.14 (0.64–7.18) | 0.216 | 1.66 (0.739-–3.77) | 0.218 |
| Good knowledge | 137 | 18 (13.1) | 1 | | 1 | | 1 | |

Note:

*Significant $p < 0.05$ (95% CI),

**Data from previous survey in 2016.

CED: Chronic Energy Deficiency, IYCF: Infant Young Children Feeding, COR: Crude Odds Ratio, AOR[a] = Adjusted Odds Ratio without LBW, P value [b] = P value without LBW, AOR[c] = Adjusted Odds Ratio with LBW, P value [d] = P value with LBW, CI = Confidence Interval, adjusted variables were age of child bearing, maternal CED risk during pregnancy, maternal dietary diversity, frequency of ANC visit and knowledge during pregnancy.

## Discussion

### Findings

This study revealed that prenatal and postnatal factors were associated with infant stunting in Sambas District, Indonesia. In model where LBW wasn't included in the multivariable analysis, preterm birth, short maternal stature, and complete immunization coverage were the key factors for the 20.8% prevalence of infant stunting in this rural area, whereas prenatal factors, including maternal CED risk during pregnancy, maternal anaemia during pregnancy, maternal dietary diversity, knowledge during pregnancy and knowledge on IYCF were not significantly associated with infant stunting in this study.

**Table 5. Household characteristics and stunting.**

| Variable | Total | Stunting | COR (95%CI) | P value | AOR (95%CI) [a] | P value [b] | AOR (95%CI) [c] | P value [d] |
|---|---|---|---|---|---|---|---|---|
| Income per capita (N = 474)** | | | | | | | | |
| <IDR 380,000 | 237 | 59 (24.9) | 1.68 (1.07–2.64) | 0.023* | 2.14 (0.49–2.58) | 0.066 | 1.53 (0.873--2.687) | 0.136 |
| ≥IDR 380,000 | 237 | 39 (16.5) | 1 | | 1 | | 1 | |
| Household food security (N = 481) | | | | | | | | |
| Food insecure | 284 | 61 (21.5) | 1.11 (0.71–1.74) | 0.655 | 1.12 (0.95–4.79) | 0.561 | 0.843 (0.475--1.49) | 0.561 |
| Food secure | 197 | 39 (19.8) | 1 | | 1 | | 1 | |

Note:

**: data from previous study in 2016.

COR: Crude Odds Ratio, AOR[a] = Adjusted Odds Ratio without LBW, P value [b] = P value without LBW, AOR[c] = Adjusted Odds Ratio with LBW, P value [d] = P value with LBW, CI = Confidence Interval, IDR = Indonesian rupiah, adjusted variables were income per capita and household food security. **ID**

In model without LBW, it was found that preterm birth was associated with stunting. This study in line with a few studies that examined the association between preterm birth and stunting, wasting and underweight in developing countries. A study conducted in Brazil reported that at the age of 1 year, the odds ratios for stunting among the late preterm infants was 2.35 [10]. These estimates are lower than our estimates (Table 3). A cohort study in rural Malawi among 840 infants found that preterm infants were at a significantly higher risk of being wasted and underweight compared with the term infants [11].

Preterm birth and intrauterine growth restrictions (IUGR) are the two underlying biological factors leading to LBW. A baby who suffered from intra-uterine growth restriction (IUGR) as a fetus was effectively born malnourished. Growth deficits since birth seemed to significantly increase the risk of stunting up until 2 years of life and contribute to a short stature as well as increasing the risk of developing chronic diseases later in life. Link between low birth weight (LBW) and child malnutrition could possibly be described by the increased vulnerability of children with LBW to infections such as diarrhea and lower respiratory infections and the increased risk of complications including sleep apnea, jaundice, anemia, chronic lung disorders, fatigue and loss of appetite compared to children with normal birth weights [12].

In multivariable analysis (model without LBW), we found that short maternal stature had association with stunting. This study result is in line with prior longitudinal study by Mendes MSF, Villamor E, and Melendez GV using the 2006 Brazilian Demographic Health Survey data that maternal height was positively associated to children's HAZ values. Children having mothers <145 cm tall have lower HAZ than those having ≥145 cm height mother. In addition, the results of those analysis also indicated the risk of child stunting by maternal height categories (<145, 145–149, 150–154, 155–159 and ≥160 cm) were 2.95 (1.51; 5.77), 2.29 (1.33; 3.93), 1.09 (0.63; 1.87), and 0.89 (0.45; 1.77), respectively. This suggests that mothers with the height of <145 cm were at greater risk of having stunted children (2.95 times) than other maternal height categories [13].

Based on a literature review and meta-analyzes conducted in 2012, maternal stature is associated with LBW owing to both genetic and environmental factors. Women with short stature may have passed relevant genes to their fetus. Short stature is also associated with smaller uterus and pelvic size, thus contributing to a smaller fetus size. An example of environmental factor that is associated with short stature is low socioeconomic status, which keeps women in a cycle of malnourishment [14].

In model where LBW included, LBW and diarrhea were associated with stunting in this study. LBW occurred due to intrauterine growth restriction during pregnancy, in which the

fetus was effectively born malnourished. Growth deficits since birth seemed to significantly increase the risk of stunting up until 2 years of life and contributed to a short stature as well as increasing the risk of developing chronic diseases later in life. The link between LBW and childhood malnutrition may be described by the increased vulnerability of children with LBW to infections such as diarrhea and lower respiratory infections and the increased risk of complications, including sleep apnea, jaundice, anemia, chronic lung disorders, fatigue, and loss of appetite, compared with children with normal birth weights [12].

LBW is closely associated with fetal and neonatal morbidity and mortality [15]. Compared with the national data collected in 2013, the proportion of LBW in the present study is much lower. In 2013, the Indonesian Ministry of Health stated that the prevalence of LBW was 10.2% [4]. Although the percentage of both LBW and diarrhea in Sambas District was lower than that on the national level, it was still a significant concern because LBW has negative impacts on infants. At population level, the proportion of infants with a LBW is an indicator of a multifaceted public health problem that includes long-term maternal malnutrition, ill health, hard work, and poor health care in pregnancy [9].

The present study showed that diarrhea was associated with stunting. The incidence of diarrhea has been shown to lead to greater risk of child stunting. Based on the findings of a study published in 2014 by the Australian Government, infection can contribute to undernutrition, which results in children being more vulnerable to infectious diseases. This finding is consistent with the framework mentioned in the literature review, which explains that when a child has an infectious disease, the malabsorption of nutrients and metabolism disturbances are also observed [16]. Another study found that frequent diarrhea in children can lead to malnutrition [17].

This study found that that >10% of infants age 0–11 months in Sambas District had acute diarrhea. This prevalence of diarrhea is higher than the rate 3.5% reported for the national prevalence in 2013 among children age <5 years [4]; however, it is important to note that the data from the Indonesian government represent an older age group. According to other studies, the prevalence of diarrhea was the highest among children age 6–11 months, remained at a high level among children age 1 year, and then decreased as the child aged [18]. A decreased prevalence of diarrhea with increased age may be due to the fact that the immune system becomes stronger with age. The peak prevalence of diarrhea occurs at age of 1–1.5 years because child starts moving around the house and begins to eat food other than breast milk [18]. Symptoms of diarrhea in children at an older age are less severe compared with those at a younger age [19]. In terms of the cause, diarrhea is significantly associated with poor sanitation [18].

In the model of logistic regression in two models (with and without LBW), complete immunization coverage of children ≥6 months was associated with stunting. This finding similar with previous studies that showed complete coverage of child immunization explains the lower prevalence of both wasting and stunting. These studies also provided evidence that a childhood vaccination program led to improved child growth in a setting in which most children were undernourished according to WHO child growth standards [20, 21]. Moreover, a study in Papua New Guinea found that children with stunting (6 children, 13.6%) were less likely to be fully vaccinated compared with children without stunting (10 children, 16.4%) (OR, 0.805; 95% CI, 0.269–2.409) [22, 23].

The study findings indicate the need for integrated interventions to reduce the stunting rate in Indonesia. Interventions should use life-cycle approaches to address various factors from the prenatal to the postnatal determinants. A strong required effort to improve the basic needs for mothers and their children. Low level of maternal knowledge related to nutrition, lack of basic immunization coverage, and the high prevalence of diarrhea should be addressed by the

local government. Health and nutrition education targeting mothers to improve healthy supplementary feeding, immunization for infants and children, and management of diarrhea among infants should be programmed and advocated with a multilevel approach involving health workers at the level of the Integrated Health Center, village midwives, and the local public health authority. Empowering women through education and economics will help improve a child's nutritional status, which can lead to better childcare practices, and improve the household economic status, which is essential for better food intake, less exposure to infections, and better use of health care services.

## Strengths and limitations

The strength of the present study is that it is a prospective study of participants enrolled in a previous study. As a result, the present study was able to capture a wide spectrum of the determinants of stunting and could assess the prenatal, perinatal, and infant periods. This study also covers 17 out of the 19 representative sub-districts in Sambas District, thus representing nearly the entire population of Sambas District. To reduce the possibility of bias and to ensure data quality, the author team trained of the researchers, conducted pretesting, used qualified enumerators, conducted daily calibration, and performed data cleaning/field checking in the field and basecamp and before data analysis. A limitation is that the statistical power of the study may have been reduced by the lost-to-follow-up rate of 10.6% (59 of 559 participants).

## Conclusion

Among pre- and postnatal conditions, several factors are significantly associated with stunting in infants 0–11 months in Sambas District, West Kalimantan, Indonesia. These factors are preterm birth, short maternal stature, and complete immunization coverage for children. In addition, these variables and the associated stunting are further affected by LBW and diarrhea. Among these factors, LBW has the strongest association with stunting. Based on these findings, the Indonesian government should place a high priority on addressing these risk factors.

## Supporting information

**S1 Questionnaire.**
(PDF)

## Acknowledgments

We would like to thank the Sambas District Office and local government for their support. We are grateful to the study participants. We also thank Trias Mahmudiono, SKM., MPH., GCAS., Ph.D (Faculty of Public Health, University of Airlangga) for his assistance and valuable discussion on the manuscript.

## Author Contributions

**Conceptualization:** Arindah Nur Sartika, Meirina Khoirunnisa, Eflita Meiyetriani, Evi Ermayani, Indriya Laras Pramesthi, Aziz Jati Nur Ananda.

**Data curation:** Eflita Meiyetriani.

**Formal analysis:** Arindah Nur Sartika, Meirina Khoirunnisa, Eflita Meiyetriani.

**Investigation:** Arindah Nur Sartika, Meirina Khoirunnisa, Evi Ermayani, Indriya Laras Pramesthi, Aziz Jati Nur Ananda.

**Methodology:** Arindah Nur Sartika, Meirina Khoirunnisa, Eflita Meiyetriani, Evi Ermayani, Indriya Laras Pramesthi, Aziz Jati Nur Ananda.

**Project administration:** Arindah Nur Sartika.

**Resources:** Arindah Nur Sartika, Meirina Khoirunnisa.

**Supervision:** Eflita Meiyetriani, Evi Ermayani, Indriya Laras Pramesthi, Aziz Jati Nur Ananda.

**Visualization:** Meirina Khoirunnisa.

**Writing – original draft:** Arindah Nur Sartika, Meirina Khoirunnisa, Eflita Meiyetriani.

**Writing – review & editing:** Eflita Meiyetriani.

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
