## [Decision Letter · Decision Letter 0]

17 Sep 2020

PONE-D-20-10544

Determinants of Stunting among Infants Aged 0-11 Months in a Borderline Area of Indonesia: Evidence from Sambas longitudinal study on Maternal and Child Nutrition, Indonesia

PLOS ONE

Dear Dr. Meiyetriani,

Thank you for submitting your manuscript to PLOS ONE. After careful consideration, we feel that it has merit but does not fully meet PLOS ONE’s publication criteria as it currently stands. Therefore, we invite you to submit a revised version of the manuscript that addresses the points raised during the review process.

We look forward to receiving your revised manuscript.

Kind regards,

Calistus Wilunda, DrPH

Academic Editor

PLOS ONE

Journal Requirements:

2. Please include additional information regarding the survey or questionnaire used in the study and ensure that you have provided sufficient details that others could replicate the analyses.

For instance, if you developed a questionnaire as part of this study and it is not under a copyright more restrictive than CC-BY, please include a copy, in both the original language and English, as Supporting Information.

In addition, please provide further details of the pre-testing of this questionnaire, for example the number of participants and where they were recruited from.

3. Please refrain from stating p values as 0.000, either report the exact value or employ the format p<0.001.

5. Please include your tables as part of your main manuscript and remove the individual files.

Please note that supplementary tables should be uploaded as separate "supporting information" files.

Additional Editor Comments:

The background section of the abstract is too long and the objective is not clear.

The article is too lengthy and there is plenty of room to make it concise.

In the methods section, please write in a continuous prose. In lines 143-147, mention the data collected first and then instrument used.

Information on the parameters finally used the estimate the sample size should be stated in the text. Additional considerations in sample size calculation may be included in a supplemental table.

The analysis should account for the survey design, particularly the cluster sampling design.

Consider including some tables as supplemental files or combining some tables.

In the Results section, present the findings without interpretation (as done for example on lines 233-234) stunting.

The entire manuscript needs to be edited for English grammar. For example, there is a mixture of present and future tense in the Methods section, which is inappropriate.

In the statistical analysis section, describe the interactions assessed and rationale behind this.

Use “multivariable analysis” instead of “multivariate analysis”.

In Table 2, combine categories with few subjects.

Consider re-categorizing “age of child bearing” as subjected in the aged <18 years are very few. Dichotomising this variable results into loss of information, so you can use more than two categories or even analyse it as a continuous variable.

In Table 13, present the adjusted OR, p value and 95% CI. The CI has upper and lower bounds, not min and max. Show the reference category as you have done in the crude analysis. State the variables adjusted for in a footnote. Ensure that you are not adjusting for variables on the causal pathway.

Clarify the study design.

Reviewers' comments:

Reviewer's Responses to Questions

**Comments to the Author**

1. Is the manuscript technically sound, and do the data support the conclusions?

Reviewer #1: Yes

Reviewer #2: Partly

Reviewer #3: Partly

2. Has the statistical analysis been performed appropriately and rigorously? 

Reviewer #1: No

Reviewer #2: Yes

Reviewer #3: Yes

3. Have the authors made all data underlying the findings in their manuscript fully available?

Reviewer #1: Yes

Reviewer #2: Yes

Reviewer #3: No

4. Is the manuscript presented in an intelligible fashion and written in standard English?

Reviewer #1: No

Reviewer #2: No

Reviewer #3: No

5. Review Comments to the Author

Reviewer #1: Thank you for the opportunity to review this paper. The study seeks to model the factors that predict stunting in younger children 0-11 months in the Sambas district, Indonesia. The study idea is useful for targeting this younger group (0-11 months) as many similar studies will focus on 6-59 month old children. But it suffers a number of methodological and reporting problems that need to be addressed. Details below.

Major comments: 1) the authors refer to the study design as longitudinal but none of the text demonstrate or support a prospective design or analysis. There was a repeat cross-sectional study at best the design used. If this is a longitudinal study, then many questions arise. There should be data on the incidence of stunting and how that was possibly measured and the use of logistic regression will not be appropriate. The authors should detail to clarify the design used.

2) While the this age group (0-11 months) may be of interest, what additional factors has this study identified which will add to the existing literature of child stunting in this group?

Additional comments:

Abstract- page 1, lines 18-20; this sentence should be revised to be clear. Page 2: lines 32-36; some factors (maternal height and preterm birth) have very wide confidence intervals including the null value. Explain why these may still be important while others were dropped from the multivariable model based on p-values and not effect estimates.

Page 2; lines 37-39: The conclusion is not very informative, revise to identify (to policy makers) which factors are important and a possible way to address them to impact stunting (in a few lines based on your results).

Background section- page 3, lines 44-45; stunting may be a problem of global concern but not a problem globally (not all settings have stunting, revise please).

Page 3, line 57; do authors mean 11.9 percent or percentage points? These are different and neither is correct as used (revise).

Page 4, lines 78-79; this point is already made.

Materials and methods- page 5, lines 91-98; include a reference to any previous studies from the bigger study that describe the methods in detail. Otherwise please provide details to clarify what design has been used. At present, a repeat cross-section is implied.

Pages 8-10; this table has no number and not referred in the text. Many definitions in the table will need a reference so may be useful to describe them in the text and provide the relevant citations. Example: definition for knowledge on stunting as presented in the table is questionable how you measured this. Classification of undernutrition (stunting etc) follow WHO definition and must be referenced.

Page 11-12; the statistical analysis section should be revised to clarify the modelling approach. If some factors were considered ‘a priori’ as potential confounders from the literature, they must be included in the model without basing their inclusion on a p-value criterion. Authors should also provide the citations to the previous studies that provided evidence for some factors to be considered potential confounders (lines 197-198).

Discussion: I think the discussion can be revised to reduce emphasis on general factors that were not associated in the present study. It is important to explain what your results mean. For example, low birth weight was a strong predictor for child stunting, what can be done about it and what shows that has any impact?

Conclusion: pages 29-30 (lines 589-618): this is a very long conclusion section which should be reduced to a few lines which summarizes your main findings and possibly recommendations arising (revise).

Tables: I think there are too many small and related tables that can be put together to improve the reading. For example, data in tables 6, 7, 8 and 9 can be combined into a single table. Table 5 can go into the supplementary materials. Tables 2 and 4 can also be combined into a single table.

Thank you.

Reviewer #2: I have read the paper with great interest. Overall the manuscript could benefit from a thorough technical editorial to ensure coherence, well summarized content and readability. The longitudinal design of the study is not clearly described. Details of the parent study are not well reflected. While details of the cross sectional survey undertaken are provided, it is difficult to draw the link between this survey conducted in 2017 and the bigger study conducted in the Sambas district. There lack of sufficient details on the follow-up between the two studies.

Some specific comments are outlined below :

Line 47: Add reference to support statement alluded to on impact of stunting on development of nations.

Line 48: Suggest adding reference on the global and regional prevalence estimates of stunting. Latest prevalence estimates are available @ https://data.unicef.org/resources/jme-report-2020/ [United Nations Children’s Fund (UNICEF), World Health Organization, International Bank for Reconstruction and Development/The World Bank. Levels and trends in child malnutrition: Key Findings of the 2020 Edition of the Joint Child Malnutrition Estimates. Geneva: World Health Organization; 2020. Licence: CC BY-NC-SA 3.0 IGO.]

Line 49 -52: Spotlight on the Americas and European region not necessary, suggest deleting.

Line 52 – 55: The discussion on insufficient progress in global stunting reductions is not well situated. Suggest the authors revise as follows: First introduce the global stunting targets, current progress – globally, regionally and in Indonesia. The WHO global tracking tool on the WHA nutrition targets can provide some useful background data https://extranet.who.int/nhdtargets/en/Stunting. Also the UNICEF, WHO, World Bank Joint Child Malnutrition Estimates (2020) edition.

Line 65-66: Rural areas in Indonesia will have a better chance at improvement under what conditions? Not clear from the way the sentence is phrased.

Line 67-69: sentence requires some editing in order to clearly convey its message.

Line 70-71: Sentence not clear. What does this sentence mean “Then from the result, there will

be a question about the baby of mothers who included in the study.”

Line 79-81: Sentence not clear.

Line 78-82: This section could benefit from re-writing to clearly communicate rationale for the study. Sentences are not well formulated.

Line 89: Materials and Methods: this paper could benefit from a brief description of the study design and population of the parent longitudinal study including the study procedures and follow-up done since it is a longitudinal design. What were the interventions provided in the parent study?

Line 109: Area and time of the study: Timeframe for the study appears very short. Was this a cross sectional survey? In line 112 – there seems to be an error in the reflection of dates for the data collection.

Line 117: which previous study. For clarity please mention. The longitudinal design of the study is not very clear and there is lack of details on the systematic follow-up period.

Line 138: were the villages chosen using probability proportional to size sampling at the first stage of sampling? What were the sampling units at the second stage of sampling? Was it any households or households with pregnant women? The sampling procedure needs to be more clearly articulated.

Line 152: How did the study address information bias especially in dealing with missing information during interviews?

Line 163: Were some of these variables (e.g Maternal Anemia during pregnancy, Maternal CED risk during Pregnancy, Knowledge during pregnancy etc) assessed from records from parent study or they were measured during this particular survey described in the paper? It is not very clear which variables were directly measured and which variables for which data was already available from previous study.

Line 186 – 192: suggest writing in past tense

Line 197: suggest highlighting the potential confounders which were considered.

Line 209 – 211: Background on Sambas district should be moved to the background or materials/methods section.

Line 256: 73.3 % children had Child Vitamin A supplementation. Is this referring to infants 6-11 months?

Line 297 – 298 : frequent episodes of wasting can also predispose infants and young children to increased risks of stunted growth.

Line 286: Discussion : I think there is scope to restructure the discussion while keeping it focused on the key findings related to the research question to prospectively evaluate the associations between pre- and postnatal conditions and stunting among infants aged 0–11 months. This should be the main focus of the discussion. Followed by discussions situating the findings within the broader global literature including other similar studies in Indonesia. From the discussion its not very clear which associations discussed are simply from bivariate analysis and which ones were found significant in multivariate analysis. This makes the discussion in its current state weak. The conclusions from this study are also not very clear.

Line 300 – 321 discusses “Exposure to Information and Knowledge”. Did the study find this to be an important determinant? Also comment applies to “ Neonatal Examination and Health Care Utilization”. Suggest focusing the discussion on the key determinants.

Reviewer #3: Title: The title is too long and its length needs to be reduced.

o For example: “Prenatal and postnatal determinants of stunting at age 0-11 months: A cross-sectional study in Indonesia”

Abstract:

• Background:

o Too long, as it includes information that should go into the methods section.

o For example, following the sentence ………. adult life, authors could simply write: “we conducted a longitudinal study on maternal and child nutrition to assess/investigate factors associated with stunting among infants aged 0-11 months in Indonesia”

• Methods:

o It is not clear how this was a longitudinal study.

o Did the study enroll pregnant women in 2016? If yes, how many and what was the gestational age at enrollment?

o Did the 2017 survey include the same mothers who were previously enrolled in the study in 2016 and infants born from those index pregnancies?

o When was the anthropometric data collected?

• Results:

o Authors should always report both the percentage and the corresponding number (n).

o Authors should also report on the key characteristics of studied mothers and children before reporting on the factors associated with stunting.

o “Furthermore, children who weighed <2.500 g at birth had the strongest

association with stunting”. Authors should consider deleting this sentence from the results section and probably put it into the conclusion, as it looks like the interpretation of the already reported result.

• Conclusion:

o This section should be improved. Authors are responsible for interpreting the key findings from this study and thereby suggest (what) interventions (by who?) to address the problems identified to be associated with stunting in infancy (0-11 months).

Manuscript:

• Methods:

o Major comment: This study was not a longitudinal study, as the outcome of interest (stunting) was only measured at one time-point. This was a cross-sectional study with some retrospective data during pregnancy.

o Authors’ calculations indicated that a sample n=492 (lines 129-135) was sufficient to conduct this study. But, was this the number of children or mothers? Was it necessary to calculate a sample size if this was a follow up survey that should include all subjects found?

o In the sampling procedure (lines 137-142), authors state that 30 villages were randomly sampled, but out of how many? Why 30 villages? Why did you have to sample?

o Instruments (lines 143-150), this should be part of the data collection procedure, and authors should avoid repetition.

o Variables and operational definition:

Minimum dietary diversity and acceptable diet – as this study only included infants aged 0-11 months, how did you categorize children under 6 months of age?

Minimum meal frequency – what was the minimum number of meals? And how did you classify under 6 months infants?

Household food security: It’s good to provide the general definition of food security, but how did authors define “household food security” in this particular study?

How was good/poor knowledge of IYCF defined?

o Data quality assurance – how was it possible to check for under- and/or over-reporting in dietary diversity data during data cleaning? Please explain!

o Data collection period – Is it April 2nd – May 17, 2017?

o Statistical analysis:

Please use the past tense.

Is it true that the multivariable model included variables with a p>0.25 in bivariate analysis? Please double-check if it was not a typo!

Please clarify the unit of analysis for this study! Is it children/mothers?

Child age and sex are factors that have been consistently found to be associated with stunting and should be forced into the final logistic regression model regardless of their p-values.

• Results:

o Lines 209-212. This paragraph should go under the “study settings” in the Methods.

o Please avoid interpretation of results in the “Results section” (example, lines 233-234). All interpretations should go under the “Discussion section”.

o It is surprising that the results’ section focused on participants’ characteristics rather than reporting on the key findings (factors associated with stunting) in line with the study objectives. Lines 281-284, please include the odds ratio to highlight the adjusted effect size of each factor on stunting.

o The number of tables (13) is more than acceptable for a research article. Please consider reducing the number of tables:

Authors could use a cross-tabulation for:

Table 2: Child characteristics and stunting

Table 3: Maternal/Caregiver characteristics and stunting

Table 4: Household characteristics and stunting

Table 5: Multivariable logistic regression model results

o Old Table 5 is not needed. A composite variable for “Good/poor knowledge of IYCF” should only be reported in the new suggested table (Table 3).

o Results’ tables should only include data on participants who were found in the 2017 survey. Therefore, the sample size should be less or equal to 500 (n≤500).

• Discussion:

o A classical discussion should focus on the key findings in terms of prenatal and postnatal factors associated with stunting in the multivariable logistic regression analysis. Authors should avoid discussing anything which is not supported by the findings of this study.

o A stunting rate of 22.5% among children aged 6-11 months was very high, and worth discussing.

o Study limitations:

Lines 583-584. What do you mean by the technical error?

Second, authors said in the methods that a sample size, n=492 was sufficient to conduct this study. Fortunately, 500 children were found for this study, so how does this become a limitation?

o Conclusions:

Conclusions should be based on the findings of this study. Authors are responsible for suggesting solutions.

6. PLOS authors have the option to publish the peer review history of their article (what does this mean?). If published, this will include your full peer review and any attached files.

Reviewer #1: **Yes: **Zakari Ali

Reviewer #2: No

Reviewer #3: **Yes: **Alphonse Nshimyiryo

---

## [Author Response · Author response to Decision Letter 0]

14 Dec 2020

Calistus Wilunda, DrPH

Academic Editor

Dear Dr. Wilunda,

Thank you for inviting us to submit a revised draft of our manuscript entitled, “Determinants of Stunting among Infants Aged 0-11 Months in a Borderline Area of Indonesia: Evidence from Sambas longitudinal study on Maternal and Child Nutrition, Indonesia” to PLOS ONE. 

Based on reviewers’ suggestions, I would like to inform you some major revision including the title and research methodology. 

a. The title of the revision manuscript: “Prenatal and postnatal determinants of stunting at age 0–11 months: A cross-sectional study in Indonesia” 

b. Research methodology: cross-sectional study was conducted as a follow-up study to assess outcomes for infants age 0–11 months by examining the pre- and postnatal conditions that may be associated with stunting.

We also appreciate the time and effort you and each of the reviewers have dedicated to providing insightful feedback on ways to strengthen our paper. Thus, it is with great pleasure that we resubmit our article for further consideration. We have incorporated changes that reflect the detailed suggestions you have graciously provided. We also hope that our edits and the responses we provide below satisfactorily address all the issues and concerns you and the reviewers have noted.

Please find enclosed the revised manuscript for further consideration and detailed reply to the reviewer comments attached with this revision. The manuscript has been revised according to the comments raised by the reviewer to the best of our ability. Again, thank you for giving us the opportunity to strengthen our manuscript with your valuable comments and queries. We have worked hard to incorporate your feedback and hope that these revisions persuade you to accept our submission in PLOS ONE.

Sincerely,

Eflita Meiyetriani

Editor’s Suggestions:

RESPONSE: Thank you for your suggestion. We have incorporated your comments by amended our manuscript using PLOS ONE's style requirements, including those for file naming. 

2. Please include additional information regarding the survey or questionnaire used in the study and ensure that you have provided sufficient details that others could replicate the analyses.

For instance, if you developed a questionnaire as part of this study and it is not under a copyright more restrictive than CC-BY, please include a copy, in both the original language and English, as Supporting Information.

In addition, please provide further details of the pre-testing of this questionnaire, for example the number of participants and where they were recruited from.

 RESPONSE: Thank you for your information. We have included a questionnaire as part of this study and it is not under a copyright more restrictive than CC-BY. This questionnaire also has been provided as Supporting Information.

3. Please refrain from stating p values as 0.000, either report the exact value or employ the format p<0.001.

RESPONSE: Thank you for your suggestion. We have amended p values as 0.000.

RESPONSE: Thank you for your information. We have included data from this study to be available in supporting information file and there are no restrictions. 

5. Please include your tables as part of your main manuscript and remove the individual files.

Please note that supplementary tables should be uploaded as separate "supporting information" files.

 RESPONSE: We thank the reviewer for this important suggestion. We have included our tables as part of our manuscript and remove the individual files. 

Additional Editor Comments:

1. The background section of the abstract is too long and the objective is not clear. 

RESPONSE: 

Thank you for your suggestion. We have amended background section. 

2. The article is too lengthy and there is plenty of room to make it concise.

 RESPONSE: 

Thank you for your suggestion. We have amended and make it concise. 

3. In the methods section, please write in a continuous prose. In lines 143-147, mention the data collected first and then instrument used.

 RESPONSE: 

We thank the reviewer for this important suggestion. We agree with you. We have amended the text where data collection first then instrument used. 

4. Information on the parameters finally used the estimate the sample size should be stated in the text. Additional considerations in sample size calculation may be included in a supplemental table.

 RESPONSE: 

We agree with you. We have revised the text and put the estimation of sample size in a supplemental table. 

5. The analysis should account for the survey design, particularly the cluster sampling design.

Consider including some tables as supplemental files or combining some tables.

In the Results section, present the findings without interpretation (as done for example on lines 233-234) stunting.

 RESPONSE: 

We have combined some tables in the manuscript and put some tables in supplemental files. Thank you. 

6. The entire manuscript needs to be edited for English grammar. For example, there is a mixture of present and future tense in the Methods section, which is inappropriate.

 RESPONSE: 

Thank you for your suggestion. We agree with you and have incorporated this suggestion throughout our paper. We have rewritten and edited the entire manuscript for English grammar including in Methods section. 

7. In the statistical analysis section, describe the interactions assessed and rationale behind this.

Use “multivariable analysis” instead of “multivariate analysis”.

 RESPONSE: 

Thank you for your insights. We have described the interaction assessment in the statistical analysis section. We have revised the sentence from multivariate analysis to multivariable analysis. 

8. In Table 2, combine categories with few subjects.

Consider re-categorizing “age of child bearing” as subjected in the aged <18 years are very few. Dichotomising this variable results into loss of information, so you can use more than two categories or even analyse it as a continuous variable.

 RESPONSE: 

Thank you for your suggestion. We agree with you that age of child bearing for aged < 18 years are very few. However, we still consider to use this category since these cut off were evaluated for our study based on previous study and literature review. 

9. In Table 13, present the adjusted OR, p value and 95% CI. The CI has upper and lower bounds, not min and max. Show the reference category as you have done in the crude analysis. State the variables adjusted for in a footnote. Ensure that you are not adjusting for variables on the causal pathway.

 RESPONSE: 

Thank you for your suggestion. We agree with you and have amended table 13. We have put the reference category in the table since we put the Crude Odd Ratio (COR) and Adjusted Odd Ratio in the same table. 

10. Clarify the study design.

 RESPONSE: 

We have clarified that the study design has been revised and changed into cross sectional study design. 

Reviewer 1 Comments:

Reviewer #1: Thank you for the opportunity to review this paper. The study seeks to model the factors that predict stunting in younger children 0-11 months in the Sambas district, Indonesia. The study idea is useful for targeting this younger group (0-11 months) as many similar studies will focus on 6-59-month-old children. But it suffers a number of methodological and reporting problems that need to be addressed. Details below.

Major comments: 

1) the authors refer to the study design as longitudinal but none of the text demonstrate or support a prospective design or analysis. There was a repeat cross-sectional study at best the design used. If this is a longitudinal study, then many questions arise. There should be data on the incidence of stunting and how that was possibly measured and the use of logistic regression will not be appropriate. The authors should detail to clarify the design used.

 RESPONSE: Thank you for your suggestion. We agree with you and have amended this suggestion throughout our paper. 

2) While the this age group (0-11 months) may be of interest, what additional factors has this study identified which will add to the existing literature of child stunting in this group?

 RESPONSE: Thank you for your question. We have reflected this comment by adding additional factors especially when the result of this study revealed significant contribution from prenatal and postnatal determinant of child stunting especially for this age group. 

Additional comments:

1. Abstract- page 1, lines 18-20; this sentence should be revised to be clear. 

 RESPONSE: 

Thank you for providing these insights. We have now abstract (p. 1, lines 18-20) has been changed. We think these changes now better. We hope that you agree.

2. Page 2: lines 32-36; some factors (maternal height and preterm birth) have very wide confidence intervals including the null value. Explain why these may still be important while others were dropped from the multivariable model based on p-values and not effect estimates.

 RESPONSE: 

You have raised an important question. There were 2 (two) reasons why we still included maternal height and preterm birth in our analysis. 1) Based on our study, we defined the variables into two categories, prenatal and postnatal factors. Maternal height was one of the prenatal factors that really important similarly with preterm birth as postnatal variable. Moreover, the result of statistical analysis shows significant relationship between preterm birth and maternal height on child stunting. 2) Since the study design has been revised into repeated cross-sectional study, we believe the effect estimates more suitable for cohort study or longitudinal study. In addition, we believe that those variables would be more appropriate to be included because the range of confidence interval were still acceptable. 

3. Page 2; lines 37-39: The conclusion is not very informative, revise to identify (to policy makers) which factors are important and a possible way to address them to impact stunting (in a few lines based on your results).

 RESPONSE: 

Thank you for your suggestion. We have redrafted the conclusion section (p. 2, lines 37-39) to establish a clearer focus.

4. Background section- page 3, lines 44-45; stunting may be a problem of global concern but not a problem globally (not all settings have stunting, revise please).

 RESPONSE: 

We agree with you. We have revised background section- page 3, lines 44-45. Thank you

5. Page 3, line 57; do authors mean 11.9 percent or percentage points? These are different and neither is correct as used (revise).

 RESPONSE: 

We agree with you. We have revised background section- page 3, line 57. It should be 10.1 %. We get the explanation for stunting reduction as follow:

Using IFLS data, Rachmi et al. (2016) reported stunting among children aged 2.0-4.9 years decreased from 50.8% in 1993 to 48.6% in 1997, 44.8% in 2000 and 36.7% in 2007.

6. Page 4, lines 78-79; this point is already made.

 RESPONSE: 

We agree with you. We have deleted page 4, lines 78-79.

7. Materials and methods- page 5, lines 91-98; include a reference to any previous studies from the bigger study that describe the methods in detail. Otherwise please provide details to clarify what design has been used. At present, a repeat cross-section is implied.

 RESPONSE: 

Thank you for your suggestion. We have revised the study design from longitudinal study to repeated cross-sectional study. We hope that the revision clarifies the points we attempted to make.

8. Pages 8-10; this table has no number and not referred in the text. Many definitions in the table will need a reference so may be useful to describe them in the text and provide the relevant citations. Example: definition for knowledge on stunting as presented in the table is questionable how you measured this. Classification of undernutrition (stunting etc) follow WHO definition and must be referenced.

 RESPONSE: 

We agree with you. We removed tables in pages 8-10 and put the definition in text and provide the relevant citation. 

9. Page 11-12; the statistical analysis section should be revised to clarify the modelling approach. If some factors were considered ‘a priori’ as potential confounders from the literature, they must be included in the model without basing their inclusion on a p-value criterion. Authors should also provide the citations to the previous studies that provided evidence for some factors to be considered potential confounders (lines 197-198).

 RESPONSE: 

We have elaborated on potential confounders (p. 11-12) and revised our model using your suggestion. We hope these revisions provide a more balanced thorough discussion. 

10. Discussion: I think the discussion can be revised to reduce emphasis on general factors that were not associated in the present study. It is important to explain what your results mean. For example, low birth weight was a strong predictor for child stunting, what can be done about it and what shows that has any impact?

 RESPONSE: 

We have rewritten discussion to be more in line with your comments. We hope that the edited section clarifies our main findings. Thank you for your suggestion. 

11. Conclusion: pages 29-30 (lines 589-618): this is a very long conclusion section which should be reduced to a few lines which summarizes your main findings and possibly recommendations arising (revise).

 RESPONSE: 

We have revised the text in the pages 29-30 lines 589-618) to reflect main findings and possibly recommendation. 

12. Tables: I think there are too many small and related tables that can be put together to improve the reading. For example, data in tables 6, 7, 8 and 9 can be combined into a single table. Table 5 can go into the supplementary materials. Tables 2 and 4 can also be combined into a single table.

Thank you.

 RESPONSE: 

We agree with you and have incorporated this suggestion throughout our paper. 

Reviewer 2 Comments: 

Reviewer #2: I have read the paper with great interest. Overall the manuscript could benefit from a thorough technical editorial to ensure coherence, well summarized content and readability. The longitudinal design of the study is not clearly described. Details of the parent study are not well reflected. While details of the cross sectional survey undertaken are provided, it is difficult to draw the link between this survey conducted in 2017 and the bigger study conducted in the Sambas district. There lack of sufficient details on the follow-up between the two studies.

 RESPONSE: 

We understand that our intention behind the study design was not clear from what was described in the

paper, and we thank the reviewer for pointing that out to us. We have revised the study design from longitudinal study to repeated cross-sectional study. We hope that the revision clarifies the points we attempted to make.

Some specific comments are outlined below:

1. Line 47: Add reference to support statement alluded to on impact of stunting on development of nations.

 RESPONSE: 

We have added a reference in line 47 which outlines to support statement alluded to on impact of stunting on development of nations. 

2. Line 48: Suggest adding reference on the global and regional prevalence estimates of stunting. Latest prevalence estimates are available @ https://data.unicef.org/resources/jme-report-2020/ [United Nations Children’s Fund (UNICEF), World Health Organization, International Bank for Reconstruction and Development/The World Bank. Levels and trends in child malnutrition: Key Findings of the 2020 Edition of the Joint Child Malnutrition Estimates. Geneva: World Health Organization; 2020. Licence: CC BY-NC-SA 3.0 IGO.]

 RESPONSE: 

Thank you for your suggestion. We have included a new reference based on your suggestion to further illustrate the global and regional prevalence estimates of stunting.

3. Line 49 -52: Spotlight on the Americas and European region not necessary, suggest deleting.

 RESPONSE: 

We agree with you and have deleted this data throughout our paper. Thank you

4. Line 52 – 55: The discussion on insufficient progress in global stunting reductions is not well situated. Suggest the authors revise as follows: First introduce the global stunting targets, current progress – globally, regionally and in Indonesia. The WHO global tracking tool on the WHA nutrition targets can provide some useful background data https://extranet.who.int/nhdtargets/en/Stunting. Also the UNICEF, WHO, World Bank Joint Child Malnutrition Estimates (2020) edition.

 RESPONSE: 

Thank you for your suggestion. We have rewritten line 52-55 and revised according your suggestion. 

5. Line 65-66: Rural areas in Indonesia will have a better chance at improvement under what conditions? Not clear from the way the sentence is phrased.

 RESPONSE: 

We have redrafted line 65-66 section to establish a clearer focus.

6. Line 67-69: sentence requires some editing in order to clearly convey its message.

 RESPONSE: 

We have revised line 67-69 section to establish a clearer sentence.

7. Line 70-71: Sentence not clear. What does this sentence mean “Then from the result, there will be a question about the baby of mothers who included in the study.”

 RESPONSE: 

We have revised line 70-71 section to establish a clearer sentence.

8. Line 79-81: Sentence not clear.

 RESPONSE: 

We have revised line 79-81 section to establish a clearer sentence.

9. Line 78-82: This section could benefit from re-writing to clearly communicate rationale for the study. Sentences are not well formulated.

 RESPONSE: 

We have rewritten this section (lines 78-82) to be more in line with your comments. We hope that the edited section clarifies the rationale of the study. 

10. Line 89: Materials and Methods: this paper could benefit from a brief description of the study design and population of the parent longitudinal study including the study procedures and follow-up done since it is a longitudinal design. What were the interventions provided in the parent study?

 RESPONSE: 

Thank you for your question. We have amended the study design using repeated cross- sectional study. Therefore, there weren’t no intervention provided both in the parent study and children study. 

11. Line 109: Area and time of the study: Timeframe for the study appears very short. Was this a cross sectional survey? In line 112 – there seems to be an error in the reflection of dates for the data collection.

 RESPONSE: 

Thank you for providing these insights. As we mentioned before, we have amended the study design. Timeframe for the study appears very short because we conducted cross sectional survey, not longitudinal study. 

12. Line 117: which previous study. For clarity please mention. The longitudinal design of the study is not very clear and there is lack of details on the systematic follow-up period.

 RESPONSE: 

We have added the explanation for previous study and changed the study design. 

13. Line 138: were the villages chosen using probability proportional to size sampling at the first stage of sampling? What were the sampling units at the second stage of sampling? Was it any households or households with pregnant women? The sampling procedure needs to be more clearly articulated.

 RESPONSE: 

We have added the explanation were the villages chosen using probability proportional to size sampling at the first stage of sampling where the second stage of sampling was households with pregnant women. 

14. Line 152: How did the study address information bias especially in dealing with missing information during interviews?

 RESPONSE: 

Thank you for your question. We have included the explanation regarding information bias in our study. This study employed several strategies to reduce information bias. We made interview guideline for the enumerator and we did pre testing questionnaire before we conducted this study. During the interview, we also asked the respondents to give some evidences regarding their answers. For example: for immunization coverage, we asked them to show their Mother and Child Health Book. Similar with birth weight, we asked them to provide the birth certificate and also the record of the history of their children. 

15. Line 163: Were some of these variables (e.g Maternal Anemia during pregnancy, Maternal CED risk during Pregnancy, Knowledge during pregnancy etc) assessed from records from parent study or they were measured during this particular survey described in the paper? It is not very clear which variables were directly measured and which variables for which data was already available from previous study. 

 RESPONSE: 

We have amended and gave explanation regarding variables that we measured. Maternal Anemia during pregnancy, Maternal CED risk during Pregnancy, Knowledge during pregnancy were assessed in the 2016 study. We called these variables as prenatal factors. LBW, Preterm birth, household economic status were measured in 2017. We called these variables as postnatal variables. 

16. Line 186 – 192: suggest writing in past tense

 RESPONSE: 

We agree with your suggestion. We have revised line 186-192 accordingly. 

17. Line 197: suggest highlighting the potential confounders which were considered.

 RESPONSE: 

We agree with your suggestion. We have revised line 197 accordingly. 

18. Line 209 – 211: Background on Sambas district should be moved to the background or materials/methods section.

 RESPONSE: 

We have put line 209-211 to the materials/methods section. 

19. Line 256: 73.3 % children had Child Vitamin A supplementation. Is this referring to infants 6-11 months?

 RESPONSE: 

Yes, we have referred children had child vitamin A supplementation to infants 6-11 months.

20. Line 297 – 298: frequent episodes of wasting can also predispose infants and young children to increased risks of stunted growth.

 RESPONSE: 

We have amended line 297-298. 

21. Line 286: Discussion: I think there is scope to restructure the discussion while keeping it focused on the key findings related to the research question to prospectively evaluate the associations between pre- and postnatal conditions and stunting among infants aged 0–11 months. This should be the main focus of the discussion. Followed by discussions situating the findings within the broader global literature including other similar studies in Indonesia. From the discussion its not very clear which associations discussed are simply from bivariate analysis and which ones were found significant in multivariate analysis. This makes the discussion in its current state weak. The conclusions from this study are also not very clear.

 RESPONSE: 

We have incorporated your comments by restructure the discussion and the key findings related to the research question. We also added explanations regarding situation on stunting in broader global literature including other similar studies in Indonesia. In our discussion, we also have revised the discussion from bivariate and multivariate analysis. We hope that the edited section makes a stronger discussion. 

22. Line 300 – 321 discusses “Exposure to Information and Knowledge”. Did the study find this to be an important determinant? Also comment applies to “ Neonatal Examination and Health Care Utilization”. Suggest focusing the discussion on the key determinants.

 RESPONSE: 

Thank you for your suggestion. We have deleted line 300-321 section. We agree that our manuscript should be focused on the key determinants for our discussion. 

Reviewer 3 Comments: 

Reviewer #3: 

1. Title: The title is too long and its length needs to be reduced.

For example: “Prenatal and postnatal determinants of stunting at age 0-11 months: A cross-sectional study in Indonesia”

 RESPONSE: 

Thank you for your suggestion, we revised the title to be more concise and precise. 

2. Abstract:

Background:

Too long, as it includes information that should go into the methods section.

For example, following the sentence ………. adult life, authors could simply write: “we conducted a longitudinal study on maternal and child nutrition to assess/investigate factors associated with stunting among infants aged 0-11 months in Indonesia”

 RESPONSE: 

Thank you for your suggestion. We revised the objective of the study into “we conducted a longitudinal study on maternal and child nutrition to assess/investigate factors associated with stunting among infants aged 0-11 months in Indonesia”.

We also condensed the background section while retaining its robust overview of the topics pertaining determinants of child stunting.

Methods:

It is not clear how this was a longitudinal study.

 RESPONSE: 

Thank you for your input. We add further explanation to clarify the concern you raised. It was repeated cross-sectional studies involving the same study participants from 2016 to 2017. We would like to asses prenatal and postnatal variables associated with stunting. 

3. Did the study enroll pregnant women in 2016? If yes, how many and what was the gestational age at enrollment?

 RESPONSE: 

The study enrolled pregnant women in 2016. Around 559 pregnant women were enrolled in the study with the gestational age at enrollment in the first trimester was 8.4 % (n=47), second trimester 37.6% (n=210) and third semester 54% (n=302)

4. Did the 2017 survey include the same mothers who were previously enrolled in the study in 2016 and infants born from those index pregnancies?

 RESPONSE: 

Yes. 2017 survey include the same mothers who were previously enrolled in the study in 2016 and infants born from those index pregnancies. 

5. When was the anthropometric data collected?

RESPONSE: The anthropometric data was collected into 2 (two) phases:

Phase 1

Maternal stature, Mid Upper Arm Circumference (MUAC) from 25 April 2016 to 11 Mei 2016. 

Phase 2

Measurement for weight and length for their babies were conducted during May 17th-April 2nd 2017. In addition, we also collected data on birth weight and birth length through questionnaire. 

6. Results:

Authors should always report both the percentage and the corresponding number (n).

RESPONSE:

 Thank you for your inputs. Please find in the revised version of our manuscript that we always report the percentage and the corresponding number

7. Authors should also report on the key characteristics of studied mothers and children before reporting on the factors associated with stunting.

RESPONSE: 

Thank you for your suggestions. We revised the results section related to the report on the key characteristics of studied mothers and children before reporting on the factors associated with stunting.

8. “Furthermore, children who weighed <2.500 g at birth had the strongest

association with stunting”. Authors should consider deleting this sentence from the results section and probably put it into the conclusion, as it looks like the interpretation of the already reported result.

RESPONSE: 

Agreed. We deleted the sentence “Furthermore, children who weighed <2.500 g at birth had the strongest association with stunting” and put it into the discussion section and conclusion. Thank you for your inputs.

9. Conclusion:

This section should be improved. Authors are responsible for interpreting the key findings from this study and thereby suggest (what) interventions (by who?) to address the problems identified to be associated with stunting in infancy (0-11 months). 

RESPONSE: 

Thank you for your suggestion. We revised the discussion who adds the stressing on the importance of our findings related to determinants of infant stunting. We believed that the low level of maternal knowledge related to nutrition, immunization and diarrhea should be addressed by the local public health centers at the sub-district level. Health and nutrition education targeting mothers to improve healthy supplementary feeding, immunization and management of diarrhea among infants should be programmed and advocated with multilevel approach involving health workers at Posyandu level, village midwife and also local public health authority.

10. Manuscript:

Methods:

Major comment: This study was not a longitudinal study, as the outcome of interest (stunting) was only measured at one time-point. This was a cross-sectional study with some retrospective data during pregnancy.

RESPONSE: 

Thank you for your inputs. We agreed with your suggestion. This research was repeated cross sectional studies. 

11. Authors’ calculations indicated that a sample n=492 (lines 129-135) was sufficient to conduct this study. But, was this the number of children or mothers? Was it necessary to calculate a sample size if this was a follow up survey that should include all subjects found?

RESPONSE: 

Thank for your questions. The initial sampling size was measured in 2016 with 559 respondents. We added description of sample size calculation and sampling technique from 2016 cross sectional study in our methodology. For the 2017 study, we agreed that all of the samples from 2016 were followed up.

12. In the sampling procedure (lines 137-142), authors state that 30 villages were randomly sampled, but out of how many? Why 30 villages? Why did you have to sample?

RESPONSE: 

Thank you for your queries. We elaborate further the sampling technique that we selected 30 clusters (villages) from total of 183 villages using PPS (Probability Proportionate to Size). 

13. Instruments (lines 143-150), this should be part of the data collection procedure, and authors should avoid repetition.

RESPONSE: 

Thank you for your suggestion. We deleted “Instruments”.

14. Variables and operational definition:

Minimum dietary diversity and acceptable diet – as this study only included infants aged 0-11 months, how did you categorize children under 6 months of age?

Response: Thank you for your question. Align with FANTA (2014), we only analyze the subsample from 6 months to 11 months. 

15. Minimum meal frequency – what was the minimum number of meals? And how did you classify under 6 months infants?

RESPONSE: 

Thank you for your question. To answer your question, we revised the method section to include the following sentence:

“In this study, minimum meal frequency is defined based on WHO (2017) as follow:

2 times for breastfed infants 6–8 months

3 times for breastfed children 9–11 months

4 times for non-breastfed children 6–11 months”

16. Household food security: It’s good to provide the general definition of food security, but how did authors define “household food security” in this particular study?

RESPONSE: 

Thank you for your question. To answer your question, we revised the method section to include the following sentence:

“In this study, household food security is defined based on FAO (2017) as follow: A state that exists when all people at all times have both physical and economic access to sufficient food to meet their dietary needs for a productive and healthy life. We categorized household food security into 2 (two) categories: a. Food Secure (0-2) b. Food Insecure (2-18)

17. How was good/poor knowledge of IYCF defined?

RESPONSE: 

Thank you for your question. To answer your question, we revised the method section to include the following sentence:

“In this study, good/poor of IYCF is defined based on median. Poor of IYCF if the score less than median, meanwhile good of IYCF if the score equal to or more than median.

18. Data quality assurance – how was it possible to check for under- and/or over-reporting in dietary diversity data during data cleaning? Please explain!

RESPONSE: 

Thank you for your concern. Our aims are to improve the quality of data collection. However, we agreed that under and over reporting seems to be unavoidable due to measurement or recall bias related to 24 hours food recall

19. Data collection period – Is it April 2nd – May 17, 2017?

Response: Yes, the data collection was divided into 2 (two) phases when April 2nd – May 17, 2017 was the second phase. 

Statistical analysis:

Please use the past tense.

RESPONSE: 

Thank you for your inputs. We will use the past tense.

20. Is it true that the multivariable model included variables with a p>0.25 in bivariate analysis? Please double-check if it was not a typo!

RESPONSE: 

Thank you for your catching this. Yes, it was a typo. It should be p < 0.25. We revised the p value throughout the manuscript. 

21. Please clarify the unit of analysis for this study! Is it children/mothers?

RESPONSE: 

Thank you for your concern. The unit analysis for this study was children. 

22. Child age and sex are factors that have been consistently found to be associated with stunting and should be forced into the final logistic regression model regardless of their p-values.

RESPONSE: 

Thank you for your suggestion. We will reanalyze the final logistic regression model to include age, sex and maternal stature.

23. Results:

Lines 209-212. This paragraph should go under the “study settings” in the Methods.

RESPONSE: 

Thank you for your suggestion. We put this paragraph under study settings.

24. Please avoid interpretation of results in the “Results section” (example, lines 233-234). All interpretations should go under the “Discussion section”.

RESPONSE: 

Thank you for your suggestion. We put the lines 233-234 under discussion section.

25. It is surprising that the results’ section focused on participants’ characteristics rather than reporting on the key findings (factors associated with stunting) in line with the study objectives. Lines 281-284, please include the odds ratio to highlight the adjusted effect size of each factor on stunting.

RESPONSE: 

Thank you for providing this insight. We put the odds ratio to highlight the adjusted effect size

The number of tables (13) is more than acceptable for a research article. Please consider reducing the number of tables:

26. Authors could use a cross-tabulation for:

Table 2: Child characteristics and stunting

Table 3: Maternal/Caregiver characteristics and stunting

Table 4: Household characteristics and stunting

RESPONSE: 

We agree with you. The number of tables has been amended. 

27. Old Table 5 is not needed. A composite variable for “Good/poor knowledge of IYCF” should only be reported in the new suggested table (Table 3).

RESPONSE: 

Thank you for your suggestion. Table 5 was deleted. 

28. Results’ tables should only include data on participants who were found in the 2017 survey. Therefore, the sample size should be less or equal to 500 (n≤500).

RESPONSE: 

Thank you for your insight. We have reflected this comment by revising the sample size using 2017 survey. 

29. Discussion:

A classical discussion should focus on the key findings in terms of prenatal and postnatal factors associated with stunting in the multivariable logistic regression analysis. Authors should avoid discussing anything which is not supported by the findings of this study.

RESPONSE: 

We agree with you. We have revised and focused on the key findings in terms of prenatal and postnatal factors associated with stunting in multivariable logistic regression analysis. 

30. A stunting rate of 22.5% among children aged 6-11 months was very high, and worth discussing.

Response: Thank you for your insight. We have reflected this comment by discussing the stunting rate among children aged 6-11 months in our manuscript. 

Study limitations:

31. Lines 583-584. What do you mean by the technical error?

RESPONSE: 

Thank you for your question. We have amended this sentence since we didn’t include macro- and micronutrient deficiencies variables in our manuscripts. 

32. Second, authors said in the methods that a sample size, n=492 was sufficient to conduct this study. Fortunately, 500 children were found for this study, so how does this become a limitation?

RESPONSE: 

Thank you for your insights. We agree with you. The text has been amended. 

33. Conclusions:

Conclusions should be based on the findings of this study. Authors are responsible for suggesting solutions. 

RESPONSE: 

You have raised an important point. We have now revising the conclusion based on the findings of this study. Thank you.

---

## [Decision Letter · Decision Letter 1]

6 Jan 2021

PONE-D-20-10544R1

Prenatal and postnatal determinants of stunting at age 0–11 months: A cross-sectional study in Indonesia

PLOS ONE

Dear Dr. Meiyetriani,

Thank you for submitting your manuscript to PLOS ONE. After careful consideration, we feel that it has merit but does not fully meet PLOS ONE’s publication criteria as it currently stands. Therefore, we invite you to submit a revised version of the manuscript that addresses the points raised during the review process.

Please use this opportunity to revise the entire manuscript for grammatical errors and clarity.

We look forward to receiving your revised manuscript.

Kind regards,

Calistus Wilunda, DrPH

Academic Editor

PLOS ONE

Additional Editor Comments (if provided):

Heading: Consider replacing “stunting at 1 age 0–11 months” with “stunting among children aged 0–11 months”

The statistical analysis section still needs to be improved.

The first paragraph under Statistical analysis is not well written and there is plenty of room for improvement. Please pay attention to the logical steps in data analysis. For example, Mantel–Haenszel analysis is usually done before multivariable logistic regression.

Line 180: What is the difference between “participant’s characteristics” and “variables included in this study”?

This sentence “To analyze normal distribution of data, the Kolmogorov–Smirnov Z test was used with a cut off point p < 0.05” does not make sense. Which variables were assessed? What is the meaning of “analyze normal distribution”?

You mention that “Mantel–Haenszel analysis was performed to control for confounding.” It is not clear which confounding variables were adjusted for. Moreover, M-H methods can only adjust for one variable. Where are results of this analysis? I think you used multivariable logistic regression to adjust for confounding. Use of M-H methods is insufficient to adjust for confounding by many factors.

Table 2 heading: replace “conditions” with “characteristics”

As mentioned in my previous comments, collapse some categories with few cases (for example ethnicity).

Table 2: Delete the subheading “Economic Condition”.

For household income and expenditure, it is unclear whether you have presented frequencies and percentages or medians. For example, this “≥2,000,000* (median)” is not clear.

Present odds ratios with 95% CIs to 2 decimal points.

Since you have indicated the exact p values, there is no need to include the footnote “*=significant p<0.05”. This footnote is redundant.

In the tables, P values of 0.000 should be written as <0.001. This has already been pointed out by one of the reviewers.

In multivariable analysis, indicate what has been adjusted for what. You can do this under statistical analysis or in the footnotes of the tables.

Note that low birth weight is an obvious mediator between maternal/household characteristics and stunting, therefore it should not be adjusted for when assessing the effect of these factors on stunting. For example, anemia during pregnancy → Low birth weight →Stunting. In this case, stunting is on the causal pathway and should not be adjusted for.

In the stunting column in the tables, you can write Stunted with the options of Yes and No. Alternatively, simply indicate the % stunted. This will make the table neater.

In the table headings, please write “Association between A and B. For example, Table 3. “Association between characteristics of children and stunting”

In the discussion, delete the subheading “Key Findings” because the text under this subheading goes beyond that.

Overall, the manuscript should be proofread to ensure it meets the journal’s English language requirements.

Reviewers' comments:

Reviewer's Responses to Questions

**Comments to the Author**

1. If the authors have adequately addressed your comments raised in a previous round of review and you feel that this manuscript is now acceptable for publication, you may indicate that here to bypass the “Comments to the Author” section, enter your conflict of interest statement in the “Confidential to Editor” section, and submit your "Accept" recommendation.

Reviewer #1: (No Response)

Reviewer #3: All comments have been addressed

2. Is the manuscript technically sound, and do the data support the conclusions?

Reviewer #1: Yes

Reviewer #3: Partly

3. Has the statistical analysis been performed appropriately and rigorously? 

Reviewer #1: Yes

Reviewer #3: Yes

4. Have the authors made all data underlying the findings in their manuscript fully available?

Reviewer #1: Yes

Reviewer #3: Yes

5. Is the manuscript presented in an intelligible fashion and written in standard English?

Reviewer #1: No

Reviewer #3: Yes

6. Review Comments to the Author

Reviewer #1: The authors have addressed most of my earlier concerns well. I have a few other minor comments for their consideration.

1. Abstract, line30; <2.500g, should be checked and written correctly (<2500g or <2.500kg).

2. Methods section, statistical analysis: The bivariate analysis are described well. Authors should add some details about the multivariable modelling performed. Did authors perform only Mantel-Haenszel (MH) analysis for controlling confounding? Clarify which factors are adjusted in the reported adjusted odds ratios. I think the MH analysis is less suitable to handle beyond 3 variables in a single model. If logistic regression was used, please describe this in the methods.

3. Page 27, line 304: references 18 and 19 need a comma separation.

Reviewer #3: Many congratulations to authors - so much effort has been made to improve this paper!! However, there are still remaining minor comments which I hope addressing them can make the paper look even better.

1. Please revise the abstract's conclusion, where you say that ONLY postnatal factors were significantly associated with stunting and none of the prenatal factors was statistically associated with stunting. In your logistic regression model, factors associated with stunting included low birth weight, a result from the intra-uterine growth restriction and it should be considered as a pre-natal factor.

2. The manuscript overall conclusion is vague - and this conclusion is not supported by the results. Authors say that several prenatal and postnatal factors were significantly associated with stunting, while only three factors (LBW, diarrhea and complete immunization) were important?

7. PLOS authors have the option to publish the peer review history of their article (what does this mean?). If published, this will include your full peer review and any attached files.

Reviewer #1: No

Reviewer #3: **Yes: **Alphonse Nshimyiryo

---

## [Author Response · Author response to Decision Letter 1]

19 Feb 2021

Dear Reviewers,

Thank you for giving us the opportunity to improve and resubmit our manuscript “PONE-D-20-10544R1: Prenatal and postnatal determinants of stunting at age 0–11 months: A cross-sectional study in Indonesia”. Please find enclosed the revised manuscript for further consideration. The manuscript has been revised according to the comments raised by the reviewer. We would like to thank the reviewer for the constructive and competent criticism, and we hope that our manuscript will be acceptable for publication in PLOS ONE. 

A.Additional Editor Comments (if provided):

B.Heading: Consider replacing “stunting at 1 age 0–11 months” with “stunting among children aged 0–11 months”

RESPONSE:

Thank you for your suggestion. We have replaced “stunting at age 0-11 months” with “stunting among children aged 0-11 months”. 

2. The statistical analysis section still needs to be improved.

RESPONSE:

Thank you for providing these insights. We have amended the statistical analysis section to be more comprehensive and clearer. 

3. The first paragraph under Statistical analysis is not well written and there is plenty of room for improvement. Please pay attention to the logical steps in data analysis. For example, Mantel–Haenszel analysis is usually done before multivariable logistic regression.

RESPONSE:

Thank you for providing these insights. As we mentioned before, we have amended the statistical analysis section including the logical steps in data analysis. We agree that Mantel-Haenszel analysis should be done before multivariable logistic regression.

4. Line 180: What is the difference between “participant’s characteristics” and “variables included in this study”?

RESPONSE:

Thank you for your question. We have put the explanation for the difference between “participant’s characteristics” and “variables included in this study” as follow:

a.“Participant’s characteristics” were variables which refers to socio-demographic variables such as: gender, educational background of the mother’s, family income and etc.

b.“Variables includes in this study” refers to prenatal and post-natal variables including: Prenatal variables: maternal short stature, maternal CED risk, knowledge during pregnancy etc. 

Post-natal variables: morbidity, birth condition and utilization of health services.

Please look at Table 1. Variables and Operational Definition, we have included participant’s characteristics in the table. 

5. This sentence “To analyze normal distribution of data, the Kolmogorov–Smirnov Z test was used with a cut off point p < 0.05” does not make sense. Which variables were assessed? What is the meaning of “analyze normal distribution”?

RESPONSE:

Thank you for raised this concern. We agree that the sentence for normal distribution should be revised. Following your suggestion, we deleted this sentence since they weren’t non parametric test that we used in this study. 

6. You mention that “Mantel–Haenszel analysis was performed to control for confounding.” It is not clear which confounding variables were adjusted for. Moreover, M-H methods can only adjust for one variable. Where are results of this analysis? I think you used multivariable logistic regression to adjust for confounding. Use of M-H methods is insufficient to adjust for confounding by many factors.

RESPONSE:

Thank you for providing these insights. We agree with you that M-H only adjust for one variable and multivariable logistic regression is suitable and correct analysis to adjust for confounding by many factors. 

7. Table 2 heading: replace “conditions” with “characteristics”

RESPONSE:

We agree with you. We have replaced “conditions” with “characteristics”.

8. As mentioned in my previous comments, collapse some categories with few cases (for example ethnicity).

RESPONSE:

Thank you for your input. We have collapsed some categories. 

9. Table 2: Delete the subheading “Economic Condition”.

RESPONSE:

We deleted the sub heading economic condition. Thank you

10. For household income and expenditure, it is unclear whether you have presented frequencies and percentages or medians. For example, this “≥2,000,000* (median)” is not clear.

RESPONSE:

Thank you for insights. We have amended household income and expenditure variables to be clear. Those variables were presented as frequencies and percentage based on median cut off. 

11. Present odds ratios with 95% CIs to 2 decimal points.

RESPONSE:

Thank you for your suggestion. We have amended odds ratios with 95% Cis to 2 decimal points. 

12. Since you have indicated the exact p values, there is no need to include the footnote “*=significant p<0.05”. This footnote is redundant.

RESPONSE:

Yes, we agree with you. We have deleted the footnote accordingly. 

13. In the tables, P values of 0.000 should be written as <0.001. This has already been pointed out by one of the reviewers.

Thank you for your input. We have revised the P values of 0.000 and written as <0.001

RESPONSE:

14. In multivariable analysis, indicate what has been adjusted for what. You can do this under statistical analysis or in the footnotes of the tables.

RESPONSE:

Thank you for your input. We have put the explanation for adjusted variables in the footnote. 

15. Note that low birth weight is an obvious mediator between maternal/household characteristics and stunting, therefore it should not be adjusted for when assessing the effect of these factors on stunting. For example, anemia during pregnancy → Low birth weight →Stunting. In this case, stunting is on the causal pathway and should not be adjusted for.

RESPONSE:

Thank you for your input. We have amended the analysis where LBW should not be adjusted when assessing the effect of these factors on stunting. 

16. In the stunting column in the tables, you can write Stunted with the options of Yes and No. Alternatively, simply indicate the % stunted. This will make the table neater.

RESPONSE:

Thank you for your suggestion. We have revised the stunting column in tables. 

17. In the table headings, please write “Association between A and B. For example, Table 3. “Association between characteristics of children and stunting”

RESPONSE:

Thank you for your input. We have amended the table headings. 

18. In the discussion, delete the subheading “Key Findings” because the text under this subheading goes beyond that.

RESPONSE:

Thank you for your input. We deleted the subheading “Key Findings”. 

19. Overall, the manuscript should be proofread to ensure it meets the journal’s English language requirements.

RESPONSE:

Thank you for your suggestion. As we stated in the first revision, we already proofread our manuscript to Enago Proofreading where the certificate was also attached in the journal platform. 

B.Reviewers' comments to the Author

Reviewer #1: The authors have addressed most of my earlier concerns well. I have a few other minor comments for their consideration.

RESPONSE:

Thank you very much for your valuable inputs and correction. We really appreciate the time and effort you and each of the reviewers have dedicated to providing insightful feedback on ways to strengthen our paper.

1. Abstract, line30; <2.500g, should be checked and written correctly (<2500g or <2.500kg).

RESPONSE:

Thank you for your input. The correct category for LBW is <2500 gram or 2.5 kilogram. 

2. Methods section, statistical analysis: The bivariate analysis are described well. Authors should add some details about the multivariable modelling performed. Did authors perform only Mantel-Haenszel (MH) analysis for controlling confounding? Clarify which factors are adjusted in the reported adjusted odds ratios. I think the MH analysis is less suitable to handle beyond 3 variables in a single model. If logistic regression was used, please describe this in the methods.

RESPONSE:

Thank you for your input. We have amended the MH analysis in our paper. We didn’t include MH analysis since we used multivariable logistic regression for adjusted more than 3 variables. 

3. Page 27, line 304: references 18 and 19 need a comma separation.

Thank you for your input. We have revised the references 18 and 19. 

Reviewer #3: Many congratulations to authors - so much effort has been made to improve this paper!! However, there are still remaining minor comments which I hope addressing them can make the paper look even better.

RESPONSE:

Thank you very much for your valuable inputs and correction. We really appreciate the time and effort you and each of the reviewers have dedicated to providing insightful feedback on ways to strengthen our paper.

1. Please revise the abstract's conclusion, where you say that ONLY postnatal factors were significantly associated with stunting and none of the prenatal factors was statistically associated with stunting. In your logistic regression model, factors associated with stunting included low birth weight, a result from the intra-uterine growth restriction and it should be considered as a pre-natal factor.

RESPONSE:

Thank you for your suggestion. We have revised the abstract’s conclusion that low birth weight, a result from the intra-uterine growth restriction and it should be considered as a pre-natal factor. 

2. The manuscript overall conclusion is vague - and this conclusion is not supported by the results. Authors say that several prenatal and postnatal factors were significantly associated with stunting, while only three factors (LBW, diarrhea and complete immunization) were important?

RESPONSE:

Thank you for your input. We have amended the manuscript specially to give more clear result from our study.

---

## [Editor Report · Decision Letter 2]

23 Feb 2021

PONE-D-20-10544R2

Prenatal and postnatal determinants of stunting among children aged 0–11 months: A cross-sectional study in Indonesia

PLOS ONE

Dear Dr. Meiyetriani,

Thank you for submitting your manuscript to PLOS ONE. Peer review for this manuscript is now complete. However, there are still minor issues that should be addressed before the manuscript can be accepted for publication.

Although you have responded that “We have put the explanation for adjusted variables in the footnote”, I could not locate this.You have also responded that “We have amended the analysis where LBW should not be adjusted when assessing the effect of these factors on stunting.” Does this mean you re-analyzed the data after excluding birthweight from the model? If this is the case, then it is surprising that the effect estimates, 95% CIS and p-values have not changed (at least this is not marked on the version with tracked changes).In line with the above, please ensure all responses to the previous comments are reflected in the revised manuscript to avoid further delays.  The manuscript still has some grammatical errors that need to be revised to meet the journal’s English language requirement.  I cannot identify all the errors, so please review the manuscript carefully. Below are examples.
The statement below under “Statistical Analysis” describes what will be done. Please change the tense to reflect what was done (note that “Categorical data will be applied...” does not make sense).

“Categorical data will be applied in a frequency table to display its frequency and percentage. Numeric data will also be reported as mean scores and standard deviations if the collected data is normally distributed. Meanwhile, if the collected data is not normally distributed, it will be presented as median and minimal-maximum value”

The heading of Table 2 is missing the word “characteristics” after “socio-demographic”For this statement “Children with low birth weight had higher odds ratio (ORs) of being stunted” it is correct to say “Children with low birth weight had higher odds of being stunted…”

We look forward to receiving your revised manuscript.

Kind regards,

Calistus Wilunda, DrPH

Academic Editor

PLOS ONE
---

## [Author Response · Author response to Decision Letter 2]

9 Apr 2021

Dear Reviewers,

Thank you for giving us the opportunity to improve and resubmit our manuscript “PONE-D-20-10544R1: Prenatal and postnatal determinants of stunting at age 0–11 months: A cross-sectional study in Indonesia”. Please find enclosed the revised manuscript for further consideration. The manuscript has been revised according to the comments raised by the reviewer. We would like to thank the reviewer for the constructive and competent criticism, and we hope that our manuscript will be acceptable for publication in PLOS ONE. 

Additional Editor Comments 

Based on editor and reviewers’ suggestions, I would like to inform you some minor revision as you mentioned in the previous email:

1.Although you have responded that “We have put the explanation for adjusted variables in the footnote”, I could not locate this.

Response: We have put the explanation for adjusted variables in the footnote. Thank you for your concern. 

2.You have also responded that “We have amended the analysis where LBW should not be adjusted when assessing the effect of these factors on stunting.” Does this mean you re-analyzed the data after excluding birthweight from the model? If this is the case, then it is surprising that the effect estimates, 95% CIS and p-values have not changed (at least this is not marked on the version with tracked changes).

Response: We have amended the analysis where LBW should not be adjusted. Thank you for reminded us. 

3.In line with the above, please ensure all responses to the previous comments are reflected in the revised manuscript to avoid further delays. 

Response: Thank you for your suggestion. We have ensured all responses were reflected in the revised manuscript.

4.The manuscript still has some grammatical errors that need to be revised to meet the journal’s English language requirement. I cannot identify all the errors, so please review the manuscript carefully. Below are examples.

The statement below under “Statistical Analysis” describes what will be done. Please change the tense to reflect what was done (note that “Categorical data will be applied...” does not make sense).

“Categorical data will be applied in a frequency table to display its frequency and percentage. Numeric data will also be reported as mean scores and standard deviations if the collected data is normally distributed. Meanwhile, if the collected data is not normally distributed, it will be presented as median and minimal-maximum value”

Response: Thank you for your suggestion. We have revised your suggestions in the revised manuscript.

5.The heading of Table 2 is missing the word “characteristics” after “socio-demographic”

Response: Thank you for your suggestion. We have revised in the revised manuscript.

6.For this statement “Children with low birth weight had higher odds ratio (ORs) of being stunted” it is correct to say “Children with low birth weight had higher odds of being stunted…”

Response: Thank you for your suggestion. We have revised in the revised manuscript.

We have incorporated changes that reflect the detailed suggestions you have graciously provided. We also hope that our edits and the responses we provide below satisfactorily address all the issues and concerns you and the reviewers have noted.

---

## [Editor Report · Decision Letter 3]

13 Apr 2021

PONE-D-20-10544R3

Prenatal and postnatal determinants of stunting among children aged 0–11 months: A cross-sectional study in Indonesia

PLOS ONE

Dear Dr. Meiyetriani,

Thank you for submitting your revised manuscript to PLOS ONE. Unfortunately, the manuscript cannot be accepted as it currently stands. Therefore, we invite you to submit a revised version of the manuscript after addressing the following minor issues.

My previous comments were not adequately addressed.  It is insufficient to say that you have revised the manuscript without stating how/where you revised or which change you made. 

Please include a clean copy and a copy with tracked changes. It is difficult to see where you made changes.

Although you have mentioned in the footnotes that you adjusted for child, maternal and household characteristics, this is too generic and will not inform the readers which specific variables you adjusted for. Although birth weight should not be adjusted for because it is an intermediate variable, it can be assessed in a separate model as an independent predictor of stunting.

The manuscript still has grammatical/spelling errors. It seems like new errors were introduced during English editing. For example, May was changed to mei (lines 99, 185), which is incorrect. Please review the entire manuscript for grammatical errors/typos. 

Change the sub-heading “Population and Subject” to “Study Population and Subjects”. In the previous version, this was written in continuous prose, which I think was a better format than the revised one.

Some statements in the methods are still in the future tense. For example this statement under the Statistical analysis section “Meanwhile, if the collected data were not normally distributed, it will be presented as median and minimal-maximum value.” Also, note that the median is usually presented with an interquartile range.

Check the tense in these sentences too: “Using pairwise deletion, the number of cases will vary. It will be the number of cases where both variables in the correlation are present”. The second sentence is not clear. What will be?

This statement is still not clear “Categorical data were applied in a frequency table to display its frequency and percentage.” You could say you summarized categorical data using frequencies and percentages, or something like that.

Use “multivariable analysis” instead of “multivariate analysis”

“A significant association was determined at a p value <0.05, at α = 5%.” What is the difference between the two values? α = 5% seems like a repetition. Consider whether you need this statement because you have stated on line 197/198 that “For all tests, a P-value < 0.05 was considered statistically significant.”

A rebuttal letter that responds to each point raised by the academic editor. You should upload this letter as a separate file labeled 'Response to Reviewers'.A marked-up copy of your manuscript that highlights changes made to the original version. You should upload this as a separate file labeled 'Revised Manuscript with Track Changes'.An unmarked version of your revised paper without tracked changes. You should upload this as a separate file labeled 'Manuscript'.

We look forward to receiving your revised manuscript.

Kind regards,

Calistus Wilunda, DrPH

Academic Editor

PLOS ONE
---

## [Author Response · Author response to Decision Letter 3]

28 Jun 2021

Calistus Wilunda, DrPH

Academic Editor

Dear Dr. Wilunda,

Thank you for inviting us to submit a revised draft of our manuscript entitled, “Prenatal and postnatal determinants of stunting at age 0–11 months: A cross-sectional study in Indonesia” to PLOS ONE. 

Based on editor and reviewers’ suggestions, I would like to inform you some minor revision as you mentioned in the previous email:

Additional Editor Comments (April 2021)

Based on editor and reviewers’ suggestions, I would like to inform you some minor revision as you mentioned in the previous email:

1.Although you have responded that “We have put the explanation for adjusted variables in the footnote”, I could not locate this.

Response: We have put the explanation for adjusted variables in the footnote. Thank you for your concern. 

2.You have also responded that “We have amended the analysis where LBW should not be adjusted when assessing the effect of these factors on stunting.” Does this mean you re-analyzed the data after excluding birthweight from the model? If this is the case, then it is surprising that the effect estimates, 95% CIS and p-values have not changed (at least this is not marked on the version with tracked changes).

Response: We have amended the analysis where multivariable analysis divided into 2 models. Model with LBW and without LBW. Thank you for reminded us. 

3.In line with the above, please ensure all responses to the previous comments are reflected in the revised manuscript to avoid further delays. 

Response: Thank you for your suggestion. We have ensured all responses were reflected in the revised manuscript.

4.The manuscript still has some grammatical errors that need to be revised to meet the journal’s English language requirement. I cannot identify all the errors, so please review the manuscript carefully. Below are examples.

The statement below under “Statistical Analysis” describes what will be done. Please change the tense to reflect what was done (note that “Categorical data will be applied...” does not make sense).

“Categorical data will be applied in a frequency table to display its frequency and percentage. Numeric data will also be reported as mean scores and standard deviations if the collected data is normally distributed. Meanwhile, if the collected data is not normally distributed, it will be presented as median and minimal-maximum value”

Response: Thank you for your suggestion. We have revised your suggestions in the revised manuscript.

5.The heading of Table 2 is missing the word “characteristics” after “socio-demographic”

Response: Thank you for your suggestion. We have revised in the revised manuscript.

6.For this statement “Children with low birth weight had higher odds ratio (ORs) of being stunted” it is correct to say “Children with low birth weight had higher odds of being stunted…”

Response: Thank you for your suggestion. We have revised in the revised manuscript.

Additional Editor Comments (May, 2021)

1.My previous comments were not adequately addressed. It is insufficient to say that you have revised the manuscript without stating how/where you revised or which change you made. 

Response: Thank you for your concern. Apologize for the mistake. I have put the wrong file in the manuscript portal. In the latest revision, I already put the manuscript with track changes. 

2.Please include a clean copy and a copy with tracked changes. It is difficult to see where you made changes.

Response: Thank you for your concern. Apologize for the mistake. I have put the wrong file in the manuscript portal. In the latest revision, I already put the manuscript with track changes.

3.Although you have mentioned in the footnotes that you adjusted for child, maternal and household characteristics, this is too generic and will not inform the readers which specific variables you adjusted for. Although birth weight should not be adjusted for because it is an intermediate variable, it can be assessed in a separate model as an independent predictor of stunting.

Response: Thank you for your suggestion. We already put the specific variables for adjusted variables that we used in the model. 

4.The manuscript still has grammatical/spelling errors. It seems like new errors were introduced during English editing. For example, May was changed to mei (lines 99, 185), which is incorrect. Please review the entire manuscript for grammatical errors/typos. 

Response: Thank you for your input. We revised the error/typos in our manuscript. 

5.Change the sub-heading “Population and Subject” to “Study Population and Subjects”. In the previous version, this was written in continuous prose, which I think was a better format than the revised one.

Response: Thank you for your input. We already revised the error/typos in our manuscript. 

6.Some statements in the methods are still in the future tense. For example this statement under the Statistical analysis section “Meanwhile, if the collected data were not normally distributed, it will be presented as median and minimal-maximum value.” Also, note that the median is usually presented with an interquartile range.

Response: Thank you for your suggestion. We already put interquartile range in our manuscripts (statistical analysis section and result section). 

7.Check the tense in these sentences too: “Using pairwise deletion, the number of cases will vary. It will be the number of cases where both variables in the correlation are present”. The second sentence is not clear. What will be?

Response: Thank you for your concern. After we elaborated the method, we revised the method. Pairwise deletion was changed to listwise deletion. We run the analysis who had complete cases. 

8.This statement is still not clear “Categorical data were applied in a frequency table to display its frequency and percentage.” You could say you summarized categorical data using frequencies and percentages, or something like that.

Response: we have revised the statement. Thank you for your input. 

9.Use “multivariable analysis” instead of “multivariate analysis”

Response: Thank you for your suggestion. We changed it from multivariate to multivariable analysis. 

10.“A significant association was determined at a p value <0.05, at α = 5%.” What is the difference between the two values? α = 5% seems like a repetition. Consider whether you need this statement because you have stated on line 197/198 that “For all tests, a P-value < 0.05 was considered statistically significant.”

Response: Thank you for your suggestion. We have revised the statement. 

We have incorporated changes that reflect the detailed suggestions you have graciously provided. We also hope that our edits and the responses we provide below satisfactorily address all the issues and concerns you and the reviewers have noted. 

Please find enclosed the revised manuscript for further consideration and detailed reply to the reviewer comments attached with this revision. The manuscript has been revised according to the comments raised by the reviewer to the best of our ability. Again, thank you for giving us the opportunity to strengthen our manuscript with your valuable comments and queries. We have worked hard to incorporate your feedback and hope that these revisions persuade you to accept our submission in PLOS ONE.

Sincerely,

Eflita Meiyetriani

Email: eflita@seameo-recfon.org

SEAMEO Regional Center for Food and Nutrition, Pusat Kajian Gizi Regional Universitas Indonesia, Jakarta, Indonesia

---

## [Editor Report · Decision Letter 4]

30 Jun 2021

Prenatal and postnatal determinants of stunting among children aged 0–11 months: A cross-sectional study in Indonesia

PONE-D-20-10544R4

Dear Dr. Meiyetriani,

We’re pleased to inform you that your manuscript has been judged scientifically suitable for publication and will be formally accepted for publication once it meets all outstanding technical requirements.

Kind regards,

Calistus Wilunda, DrPH

Academic Editor

PLOS ONE
---

## [Editor Report · Acceptance letter]

5 Jul 2021

PONE-D-20-10544R4 

Prenatal and postnatal determinants of stunting at age 0–11 months: A cross-sectional study in Indonesia 

Dear Dr. Meiyetriani:

I'm pleased to inform you that your manuscript has been deemed suitable for publication in PLOS ONE. Congratulations! Your manuscript is now with our production department. 

Kind regards, 

on behalf of

Dr. Calistus Wilunda 

Academic Editor

PLOS ONE